# Towards Assessing and Benchmarking Risk-Return Tradeoff of Off-Policy Evaluation

**Haruka Kiyohara**[*]
Cornell University

**Ren Kishimoto**[*]
Tokyo Institute of Technology

**Kosuke Kawakami**
HAKUHODO Technologies Inc.

**Ken Kobayashi**
Tokyo Institute of Technology

**Kazuhide Nakata**
Tokyo Institute of Technology

**Yuta Saito**
Cornell University

## Abstract

**Off-Policy Evaluation (OPE)** aims to assess the effectiveness of counterfactual policies using only offline logged data and is often used to identify the top-$k$ promising policies for deployment in online A/B tests. Existing evaluation metrics for OPE estimators primarily focus on the "accuracy" of OPE or that of downstream policy selection, neglecting risk-return tradeoff in the subsequent online policy deployment. To address this issue, we draw inspiration from portfolio evaluation in finance and develop a new metric, called **SharpeRatio@k**, which measures the risk-return tradeoff of policy portfolios formed by an OPE estimator under varying online evaluation budgets ($k$). We validate our metric in two example scenarios, demonstrating its ability to effectively distinguish between low-risk and high-risk estimators and to accurately identify the most *efficient* one. Efficiency of an estimator is characterized by its capability to form the most advantageous policy portfolios, maximizing returns while minimizing risks during online deployment, a nuance that existing metrics typically overlook. To facilitate a quick, accurate, and consistent evaluation of OPE via SharpeRatio@k, we have also integrated this metric into an open-source software, **SCOPE-RL**[1]. Employing SharpeRatio@k and SCOPE-RL, we conduct comprehensive benchmarking experiments on various estimators and RL tasks, focusing on their risk-return tradeoff. These experiments offer several interesting directions and suggestions for future OPE research.

## 1 Introduction

Reinforcement Learning (RL) has achieved considerable success in a variety of applications requiring sequential decision-making. Nonetheless, its online learning approach is often seen as problematic due to the need for active interaction with the environment, which can be risky, time-consuming, and unethical (Fu et al., 2021; Matsushima et al., 2021). To mitigate these issues, learning new policies offline from existing historical data, known as Offline RL (Levine et al., 2020), is becoming increasingly popular for real-world applications (Qin et al., 2021). Typically, in the offline RL lifecycle, promising candidate policies are initially screened through Off-Policy Evaluation (OPE) (Fu et al., 2020), followed by the selection of the final production policy from the shortlisted candidates using more dependable online A/B tests (Kurenkov & Kolesnikov, 2022), as shown in Figure 1.

When evaluating the efficacy of OPE methods, research has largely concentrated on "accuracy" metrics like mean-squared error (MSE) (Uehara et al., 2022; Voloshin et al., 2019), rank correlation (rankcorr) (Fu et al., 2021; Paine et al., 2020), and regret in subsequent policy selection (Doroudi et al., 2017; Tang & Wiens, 2021). However, these existing metrics do not adequately assess the balance between risk and return experienced by an estimator during the online deployment of selected policies. Crucially, MSE and rankcorr fall short in distinguishing whether an estimator is underevaluating near-optimal policies or overevaluating poor-performing ones, which influence the risk-return dynamics in OPE and policy selection in different ways. The regret metric solely emphasizes the optimal outcome

---

[*]This work was done during their internship at negocia, Inc. Correspondence to: hk844@cornell.edu

[1]**https://github.com/hakuhodo-technologies/scope-rl**

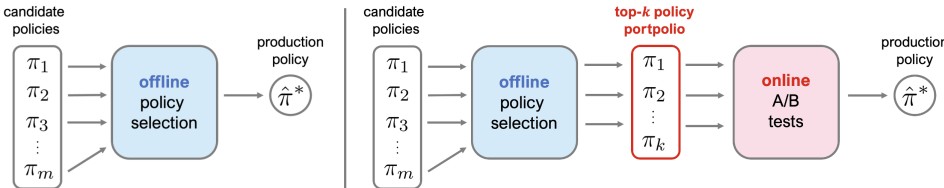

Figure 1: (Left) Conventional (off-)policy selection directly chooses the production policy via OPE. (Right) Practical workflow of policy evaluation and selection involves OPE as a screening process where an OPE estimator ($\hat{J}$) chooses top-$k$ candidate policies that are to be tested in online A/B tests, where $k$ is a pre-defined online evaluation budget. A policy that is identified as the best policy based on the evaluation process will be chosen as the production policy ($\hat{\pi}^*$).

among the top-$k$ policies selected by OPE, neglecting the potential adverse effects on the environment when implementing suboptimal policies in online A/B testing scenarios.

**Contributions.** Aiming for a more informative evaluation-of-OPE, we propose a new evaluation-of-OPE metric called **SharpeRatio@k**. This metric evaluates both the potential risk and return when deploying the resulting top-$k$ policies in the A/B testing phase. Drawing inspiration from financial portfolio management, we view the shortlisted candidate policies as a *policy portfolio* formulated by an OPE estimator. This approach allows us to assess the risk, return, and *efficiency* of an estimator based on the statistics of its policy portfolio. Specifically, we use the Sharpe ratio (Sharpe, 1998) to measure OPE *efficiency*. A policy portfolio is deemed *efficient* if it includes policies that substantially enhance the performance of the behavior (data collection) policy (*high return*), while avoiding inclusion of underperforming policies that could detrimentally affect rewards during A/B testing (*low risk*). As an initial validation, we showcase in two example scenarios how SharpeRatio@k can discern between high- and low-risk OPE estimators, a distinction not captured by any existing metric.

In addition to developing SharpeRatio@k, we have also implemented this metric in an open-source software named SCOPE-RL (Kiyohara et al., 2023). SCOPE-RL serves as a versatile tool and testing platform for OPE estimators, providing user-friendly APIs and comprehensive insights into the risk-return tradeoff of OPE estimators. Using SharpeRatio@k and SCOPE-RL, we conduct extensive benchmark experiments with various OPE estimators and RL environments, evaluating their risk-return tradeoffs across different online evaluation budgets (i.e., varying $k$ values).

Our contributions can be summarized as follows:

- **New Evaluation-of-OPE Metric:** We have introduced SharpeRatio@k, a novel evaluation metric specifically designed to assess the risk, return, and efficiency of OPE estimators, offering a more comprehensive analysis regarding the evaluation of OPE.

- **Benchmarks and Future Directions:** Through example scenarios and a range of RL tasks, we demonstrate that SharpeRatio@k is capable of measuring the risk, return, and efficiency of estimators whereas existing metrics fail to do so. We also suggest several future directions for OPE research based on the results that SharpeRatio@k produces.

- **Open-Source Implementation:** We have developed and integrated SharpeRatio@k into an open-source software named SCOPE-RL. This software is tailored to enable quick, accurate, and insightful evaluation of OPE, enhancing the accessibility and utility of our metric.

## 2    OFF-POLICY EVALUATION IN RL

We consider a general RL setup, formalized by a Markov Decision Process (MDP) defined by the tuple $\langle \mathcal{S}, \mathcal{A}, \mathcal{T}, P_r, \gamma \rangle$. Here, $\mathcal{S}$ represents the state space and $\mathcal{A}$ denotes the action space, which can either be discrete or continuous. Let $\mathcal{T} : \mathcal{S} \times \mathcal{A} \to \mathcal{P}(\mathcal{S})$ be the state transition probability, where $\mathcal{T}(s'|s, a)$ is the probability of observing state $s'$ after taking action $a$ in state $s$. $P_r : \mathcal{S} \times \mathcal{A} \times \mathbb{R} \to [0, 1]$ represents the probability distribution of the immediate reward, and $R(s, a) := \mathbb{E}_{r \sim P_r(r|s,a)}[r]$ is the expected immediate reward when taking action $a$ in state $s$. $\pi : \mathcal{S} \to \mathcal{P}(\mathcal{A})$ denotes a *policy*, where $\pi(a|s)$ is the probability of taking action $a$ in state $s$.

Off-Policy Evaluation (OPE) aims to evaluate the expected reward under an evaluation (new) policy (called the *policy value* of an evaluation policy), using only a fixed logged dataset. More precisely, the policy value is defined as the following expected trajectory-wise reward obtained by deploying an evaluation policy $\pi$:

$$J(\pi) := \mathbb{E}_{\tau \sim p_\pi(\tau)} \left[ \sum_{t=0}^{T-1} \gamma^t r_t \right],$$

where $\gamma \in (0, 1]$ is a discount factor and $p_\pi(\tau) = p(s_0) \prod_{t=0}^{T-1} \pi(a_t|s_t)\mathcal{T}(s_{t+1}|s_t, a_t)P_r(r_t|s_t, a_t)$ is the probability of observing a trajectory under evaluation policy $\pi$. The logged dataset $\mathcal{D}$ available for OPE is generated by a behavior policy $\pi_b$ (which is different from $\pi$) as

$$\mathcal{D} := \{(s_t, a_t, s_{t+1}, r_t)\}_{t=0}^{T-1} \sim p(s_0) \prod_{t=0}^{T-1} \pi_b(a_t|s_t)\mathcal{T}(s_{t+1}|s_t, a_t)P_r(r_t|s_t, a_t).$$

Given the logged dataset $\mathcal{D}$, the typical goal of OPE is to develop an estimator $\hat{J}(\pi; \mathcal{D})$ that can accurately estimate the true value of $\pi$, i.e., $J(\pi)$. The complexity of OPE arises from the fact that we only have access to logged feedback for actions selected by the behavior policy. This necessitates tackling challenges like *counterfactual estimation* and *distribution shift*, both of which can significantly contribute to high bias and variance in estimations (Liu et al., 2018; Thomas et al., 2015) (Appendix A.1 provides a detailed overview of some notable OPE estimators developed to address these specific challenges.).

**Practical workflow of policy evaluation and selection.** Due to the presence of bias-variance issues, we cannot exclusively depend on OPE results for selecting a policy for production. Instead, a more practical approach often involves a combination of OPE results and online A/B testing, offering a balance of speed and reliability in policy evaluation and selection (Kurenkov & Kolesnikov, 2022). Specifically, as demonstrated in Figure 1 (Right), this practical approach starts with the elimination of harmful policies using OPE results, followed by A/B testing of the top-$k$ shortlisted policies for a more dependable online assessment. In the following sections, we focus on this practical workflow, aiming to evaluate the effectiveness of OPE estimators in choosing the top-$k$ candidate policies for further assessment in online A/B testing.

**Existing evaluation-of-OPE metrics and their issues.** In existing literature, the following three metrics are often used to evaluate and compare the "accuracy" of OPE estimators:

- **Mean Squared Error (MSE)** (Voloshin et al., 2019): This metric measures the estimation accuracy of estimator $\hat{J}$ among a set of policies $\Pi$ as $(1/|\Pi|) \sum_{\pi \in \Pi} \mathbb{E}_\mathcal{D}[(\hat{J}(\pi; \mathcal{D}) - J(\pi))^2]$.

- **Rank Correlation (Rankcorr)** (Fu et al., 2021; Paine et al., 2020): This metric measures how well the ranking of candidate policies is preserved in the OPE results and is defined as the spearman's rank correlation between $\{J(\pi)\}_{\pi \in \Pi}$ and $\{\hat{J}(\pi; \mathcal{D})\}_{\pi \in \Pi}$.

- **Regret@$k$** (Doroudi et al., 2017): This metric measures how well the best policy among the top-$k$ candidate policies selected by an estimator performs. In particular, Regret@1 measures the performance difference between the true best policy $\pi^*$ and the best policy estimated by the estimator as $J(\pi^*) - J(\hat{\pi}^*)$ where $\hat{\pi}^* := \arg\max_{\pi \in \Pi} \hat{J}(\pi; \mathcal{D})$.

MSE measures the accuracy of OPE estimation whereas the latter two metrics assess the accuracy of policy selection. By combining these metrics, we can quantify how likely an OPE estimator can choose a near-optimal policy in policy selection (Figure 1 (Left)). However, a significant limitation of these metrics is their failure to assess the potential *risk* associated with implementing harmful policies, a frequent occurrence in more practical two-stage selection workflow having online A/B tests as a final process (Figure 1 (Right)). For example, in the scenario depicted in Figure 2, all three metrics yield identical evaluations for both estimators X and Y. Yet, with estimator X underestimating the value of near-optimal policies and estimator Y overestimating that of detrimental policies, there is a marked difference in their risk-return tradeoff, which existing metrics overlook by design. This discrepancy underscores the need for a new evaluation-of-OPE metric that can accurately quantify an estimator's risk-return tradeoff, going beyond mere accuracy assessment.

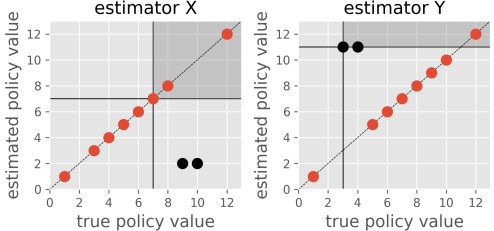

Figure 2: A toy example illustrating the situation where existing metrics (MSE, Rankcorr, and Regret) fail to evaluate the risk in the off-policy selection (OPS) task. Both estimators X and Y conduct OPS on the same set of candidate policies. X underestimates the values of the policies indicated by the black dots while Y overestimates them. The shaded regions show the top-3 policies (policy portfolio) selected by each estimator, indicating that Y is riskier than X since Y includes worse policies in its policy portfolio. Nonetheless, existing metrics give completely identical evaluations for X and Y.

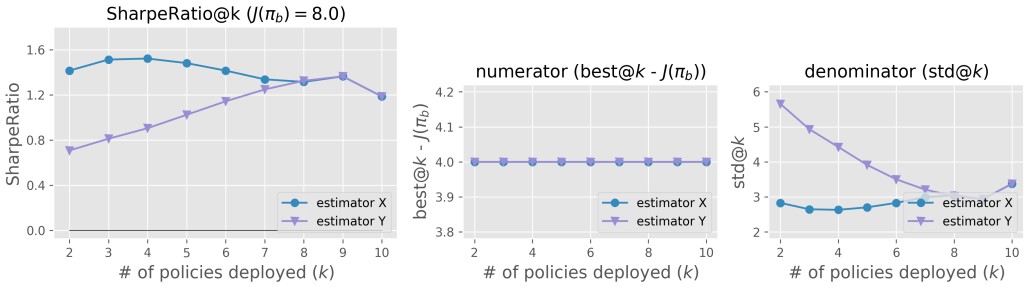

Figure 3: Evaluating estimators X and Y in the toy example of Figure 2 with SharpeRatio@k.

# 3   EVALUATING THE RISK-RETURN TRADEOFF IN OPE VIA SHARPERATIO@K

Driven by the inadequacy in assessing risk-return tradeoff in existing literature, we introduce a novel evaluation metric for OPE estimators, which we call SharpeRatio@k. This evaluation-of-OPE metric conceptualizes the top-$k$ candidate policies selected by an estimator as its "policy portfolio" inspired by financial risk-return assessments (Connor et al., 2010; Sharpe, 1998).

> **SharpeRatio@k** is the proposed metric formulated to assess the risk-return tradeoff of an OPE estimator, considering a predefined online evaluation budget $k$. It is defined as follows:
>
> $$\textbf{SharpeRatio@k}(\hat{J}) := \frac{\text{best@}k(\hat{J}) - J(\pi_b)}{\text{std@}k(\hat{J})},$$
>
> where best@$k(\hat{J})$ represents the value of the highest-performing policy within the top-$k$ policies under the estimator $\hat{J}$ (the policy portfolio of $\hat{J}$), and std@$k(\hat{J})$ refers to the standard deviation of policy values among these top-$k$ policies selected by the estimator. These are defined more precisely as:
>
> $$\text{best@}k(\hat{J}) := \max_{\pi \in \Pi_k(\hat{J})} J(\pi), \;\; \text{std@}k(\hat{J}) := \sqrt{\frac{1}{k} \sum_{\pi \in \Pi_k(\hat{J})} \left( J(\pi) - \left( \frac{1}{k} \sum_{\pi \in \Pi_k(\hat{J})} J(\pi) \right) \right)^2},$$
>
> where $\Pi_k(\hat{J})$ denotes the set of top-$k$ policies as determined by the estimated policy values under estimator $\hat{J}$ (i.e., the policy portfolio formed by $\hat{J}$).

Note that we include the behavior policy $\pi_b$ as one of the candidate policies when computing SharpeRatio@k, and thus it is always non-negative and behaves differently given different $\pi_b$.

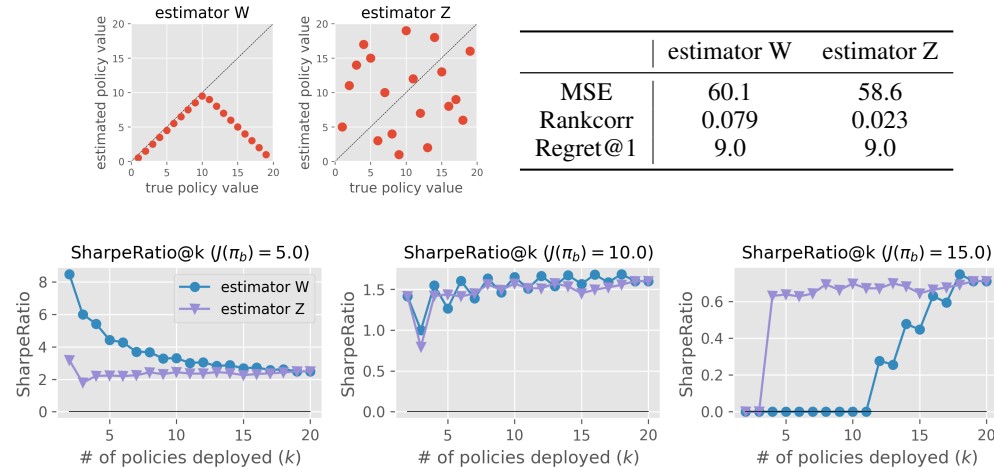

Figure 4: A toy example illustrating the case of evaluating a conservative OPE (estimator W) and uniform random selection (estimator Z) with conventional evaluation-of-OPE metrics (the right top table) and SharpeRatio@k (the bottom figures).

Our SharpeRatio@k calculates the return (best@$k$) relative to a risk-free baseline ($J(\pi_b)$), while incorporating risk (std@$k$) in its denominator.[2] Reporting SharpeRatio@k under varying online evaluation budgets, i.e., different values of $k$, is particularly useful to evaluate and understand the risk-return tradeoff of OPE estimators. Below, we showcase how SharpeRatio@k provides valuable insights for comparing OPE estimators in two practical scenarios while the existing metrics fall short.

***Scenario 1: Overestimation vs. Underestimation.*** The first example is the one illustrated in Figure 2, where current evaluation-of-OPE metrics fail to assess the risk of overestimating the effectiveness of underperforming policies, producing identical values for estimators X and Y. In this context, SharpeRatio@k offers more valuable insights. As evident in Figure 3, it assigns a higher score to estimator X than to Y. To delve into the mechanisms of SharpeRatio@k, we have separately charted its numerator (return) and denominator (risk) in Figure 3. This breakdown of SharpeRatio@k reveals that the return (best@$k(\hat{J}) - J(\pi_b)$) is identical for both X and Y, but the risk (std@$k(\hat{J})$) is substantially greater for estimator Y. This is because Y overestimates the value of underperforming policies, thereby increasing the risk of implementing these harmful policies in future online A/B tests. Consequently, in this particular instance, estimator X is a better choice than Y, a distinction that existing metrics utterly fail to recognize.

***Scenario 2: Conservative vs. Random.*** In our next example, we compare a conservative OPE estimator (estimator W, which consistently underestimates) with a uniform random estimator (estimator Z), as depicted in Figure 4. Similarly to the prior example, the table in Figure 4 indicates that conventional metrics yield nearly the same values for W and Z, despite their distinctly different behaviors. This obscures the selection of the more suitable estimator. SharpeRatio@k excels in identifying a more effective estimator, considering the risk-return tradeoff and the specific problem instance. Specifically, Figure 4 demonstrates how SharpeRatio@k evaluates estimators W and Z under three different behavior policies, each with varying effectiveness ($J(\pi_b) = 5.0, 10, 15$, where higher values signify more effective behavior policy $\pi_b$). The results show that when $\pi_b$ is less effective ($J(\pi_b) = 5.0$), SharpeRatio@k favors estimator W for its lower risk. Conversely, with a moderately effective $\pi_b$ ($J(\pi_b) = 10$), SharpeRatio@k shows no preference between the two estimators, suggesting their similar efficiencies in this context. Lastly, in scenarios where $\pi_b$ is highly effective ($J(\pi_b) = 15$), estimator Z is preferred by SharpeRatio@k due to its higher potential for selecting superior policies compared to $\pi_b$. Thus, SharpeRatio@k proves to be a more informative and practical metric than conventional accuracy metrics, providing crucial insights for selecting the most efficient estimator in relation to their risk-return tradeoff.

---

[2]By subtracting the baseline in the numerator, we can avoid an undesired edge case where an estimator that chooses only policies that are worse than the behavior policy but accidentally has a small std is evaluated highly.

**Open-Source Implementation.** In addition to developing SharpeRatio@k, to facilitate its concise and precise usage in future research and practice, we provide its implementation and examples in SCOPE-RL, an open-source python software for offline RL and OPE (Kiyohara et al., 2023). Our experiments in the following section are also implemented on top of this software.

## 4 BENCHMARKING OPE ESTIMATORS WITH SHARPERATIO@K

This section evaluates typical OPE estimators with SharpeRatio@k and SCOPE-RL, which results in several valuable future directions and recommendations for OPE research.[3]

### 4.1 EXPERIMENT SETTING

**RL tasks.** We use a total of 7 tasks including Reacher, InvertedPendulum, Hopper, Swimmer (continuous control) from Gym-Mujuco (Brockman et al., 2016; Todorov et al., 2012) and CartPole, MountainCar, Acrobot (discrete control) from Gym-Classic Control (Brockman et al., 2016). Below, we describe the experiment setting for continuous control tasks in detail. A similar procedure is applicable to the discrete control tasks as described in Appendix A.2.

**Data Generation.** We initially train a behavior policy using SAC (Haarnoja et al., 2018) in an online environment and then introduce Gaussian noise to induce stochasticity, creating a logged dataset $\mathcal{D}^{(tr)}$ for the offline training of candidate policies. To establish a diverse set of candidate policies ($\Pi$), we train CQL (Kumar et al., 2020) and IQL (Kostrikov et al., 2022), each implemented in d3rlpy (Seno & Imai, 2021), using three distinct neural networks for each method. By adding Gaussian noise with four different standard deviations to each of these trained policies, we generate a total of 24 policies. From this assortment, we randomly select 10 policies for each experiment to form our set of candidate policies ($\Pi$). We conduct OPE simulations based on logged dataset $\mathcal{D}^{(te)}$ with $n = 10,000$ trajectories generated under the behavior policy and 10 different random seeds. Note that we report MSE and Regret normalized by the average and maximum policy value among candidate policies, respectively as nMSE and nRegret@1.

### 4.2 COMPARED ESTIMATORS

We evaluate the following estimators using SharpeRatio@k and conventional accuracy metrics. Appendix A.1 provides rigorous definitions and detailed descriptions of these estimators.

**Direct Method (DM)** (Le et al., 2019): This estimator approximates the Q-function (state-action value) $\hat{Q}(s_t, a_t) \approx R(s_t, a_t) + \mathbb{E}[\sum_{t'=t+1}^{T-1} \gamma^{t'-t} r_t | s_t, a_t, \pi]$ using the logged data and then estimates the policy value based on this estimated Q-function. This estimator can produce high bias when $\hat{Q}$ is inaccurate, while its variance is often smaller than other methods.

**Per-Decision Importance Sampling (PDIS)** (Precup et al., 2000): Leveraging the sequential structure of the MDP, this estimator corrects the distribution shift between policies by multiplying the importance weights of previous actions at each time step $t$, i.e., $\prod_{t'=0}^{t} \frac{\pi(a_{t'}|s_{t'})}{\pi_b(a_{t'}|s_{t'})}$. This estimator is unbiased, but it suffers from high variance, especially when $T$ is large.

**Doubly Robust (DR)** (Jiang & Li, 2016; Thomas & Brunskill, 2016): This estimator combines model-based and importance sampling-based approaches. Specifically, it uses DM as a baseline estimation and applies importance sampling only to the residual of the discounted reward. In this way, DR often reduces variance comapred to PDIS while remaining unbiased.

**Marginal Importance Sampling (MIS)** (Liu et al., 2018; Uehara et al., 2020): This estimator relies on the marginalized importance weight ($\frac{d^\pi(s,a)}{d^{\pi_b}(s,a)}$) instead of the per-decision importance weights used in PDIS. MIS reduces variance compared to PDIS because the marginal importance weights do not depend on the trajectory length $T$. **Marginal Doubly Robust (MDR)** is its DR variant.

---

[3]https://github.com/hakuhodo-technologies/scope-rl/tree/main/experiments

Table 1: **Spearman's rank correlation in estimator ranking** and **disagreement in best estimator selection** between **SharpeRatio@5** and conventional metrics.

| metric | Reacher | Inv.Pendulum | Hopper | Swimmer | CartPole | MountainCar | Acrobot |
|---|---|---|---|---|---|---|---|
| **RankCorr** | **0.81** (7/10) | 0.18 (5/10) | **0.70** (0/10) | **0.79** (3/10) | **0.71** (10/10) | **0.57** (1/10) | **0.38** (10/10) |
| **nRegret** | **0.33** (9/10) | 0.02 (9/10) | **0.45** (3/10) | **0.45** (10/10) | **0.57** (9/10) | **-0.77** (10/10) | -0.10 (9/10) |
| **nMSE** | **0.76** (9/10) | -0.11 (8/10) | **0.83** (0/10) | 0.06 (4/10) | **0.45** (1/10) | **-0.20** (10/10) | -0.08 (10/10) |

*Note*: The value outside and inside the parentheses represent the mean of Spearman's rank correlation regarding the ranking of estimators, and the number of trials in which SharpeRatio@5 and other metrics disagree regarding best estimator selection, respectively, calculated over 10 random seeds. The **blue** font indicates instances where SharpeRatio@5 demonstrates a high correlation, characterized by the condition (mean - std > 0) where std is the standard deviation of rank correlation. Conversely, the **red** font signifies the opposite scenario, where the condition (mean + std < 0) applies.

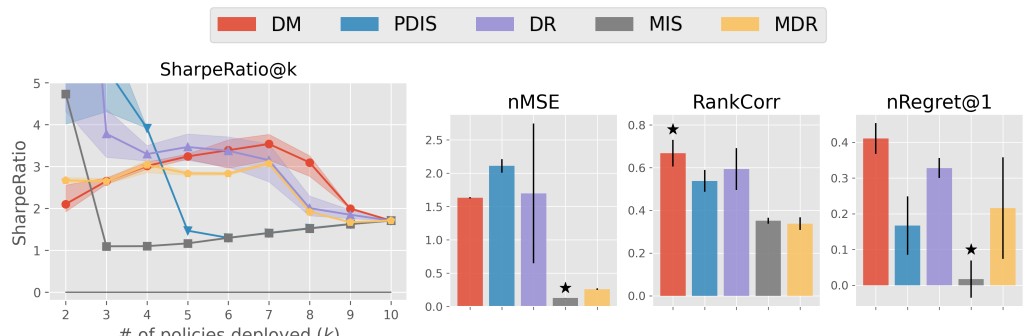

Figure 5: Evaluation-of-OPE results based on **SharpeRatio@k** (the left figure) and **conventional metrics including nMSE, RankCorr, and nRegret@1** (the right three figures) in **MountainCar**. A lower value of nMSE and nRegret@1, and a higher value of RankCorr and SharpeRatio@k mean a better estimator. The stars (⋆) indicate the best estimator under each metric.

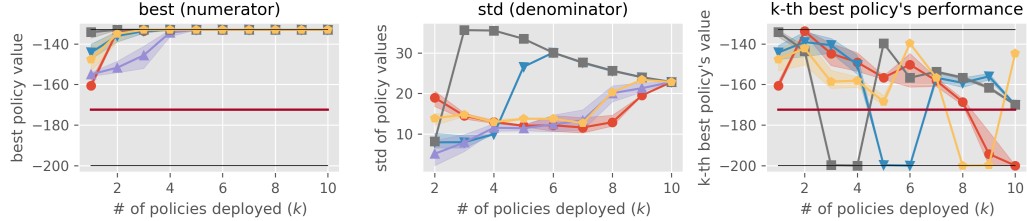

Figure 6: **Reference statistics of the top-$k$ policy portfolio** formed by each estimator in **MountainCar**. "best" is used as the numerator of SharpeRatio@k, while "std" is used as its denominator. A higher value is better for "best" and "$k$-th best policy's performance", while a lower value is better for "std". The dark red lines show $J(\pi_b)$, which is the risk-free baseline of SharpeRatio@k.

Note that we use self-normalized importance weights for PDIS, DR, MIS, and MDR to stabilize their estimations and estimate the marginal importance weights by BestDICE (Yang et al., 2020). For DM, DR, and MDR, we use FQE (Le et al., 2019) with a neural network to estimate $Q(s_t, a_t)$.

## 4.3 RESULTS

This section discusses key observations in our benchmark experiments. Appendix A.3 provides more comprehensive results and discussion with more RL environments and metrics.

***How does SharpeRatio@k differ from conventional metrics?*** Table 1 presents the Spearman's rank correlation between SharpeRatio@5 and conventional metrics (RankCorr, nRegret, and nMSE). The findings reveal that SharpeRatio@5 correlates with conventional metrics in most of the tasks

we tested, notably Reacher, Hopper, Swimmer, and CartPole. However, better estimators under conventional metrics (higher RankCorr and lower nRegret and nMSE) do not necessarily equate to higher SharpeRatio@5 values in other environments (i.e., InvertedPendulum and MountainCar). A significant deviation of SharpeRatio@5 from other metrics is particularly observed with nMSE and nRegret. This divergence is understandable, as nMSE focuses exclusively on the estimation accuracy of OPE without considering the effectiveness of policy selection, while nRegret measures only the performance of the best policy, neglecting the risks of other selected policies. In contrast, SharpeRatio@5 exhibits some correlation with RankCorr, which assesses the alignment of policy performances. However, the optimal estimator identified by SharpeRatio@5 often differs from that chosen by RankCorr, due to the fact that it can overevaluate the effectiveness of estimators that may choose harmful policies in the top-$k$ policy selection. In fact, SharpeRatio@5 and RankCorr do not agree on the best estimator in more than 50% of trials (36 trials out of 70), followed by 42 trials (out of 70) for nMSE and 59 trials (out of 70) for nRegret, across 7 different tasks and 10 random seeds as shown in the parentheses in Table 1. The following section will investigate a specific environment, MountainCar, where SharpeRatio@k and conventional metrics diverge substantially.

***SharpeRatio@k is a more appropriate metric than conventional ones.*** This section delves into a qualitative analysis using the *MountainCar* example, where SharpeRatio demonstrates a relatively weak correlation to RankCorr and displays an inverse conclusion with nMSE and nRegret. Figure 5 contrasts the benchmark results obtained with SharpeRatio@k under various online evaluation budgets $k$ against those derived from conventional metrics. Figure 6 provides reference statistics for the top-$k$ policy portfolios generated by each estimator. Importantly, the performance of the "k-th best policy" showcases how well the policy, ranked k-th by each estimator, performs. These findings illustrate that the preferred OPE estimator varies significantly depending on the evaluation metrics used. For example, MSE and Regret identify MIS as the best estimator, whereas RankCorr and SharpeRatio@7 favor DM, and SharpeRatio@4 selects PDIS. A closer examination of these three estimators, through the reference statistics in Figure 6, reveals that conventional metrics tend to ignore the risks associated with OPE estimators that mistakenly include suboptimal policies in their portfolios. Specifically, nMSE and nRegret overlook the risk of MIS being a worst-case estimator for $k \geq 3$. Moreover, RankCorr fails to recognize the risk of PDIS being a near worst-case estimator for $k \geq 5$, and it incorrectly ranks PDIS higher than MDR, which avoids deploying a suboptimal policy until $k = 8, 9$. In contrast, SharpeRatio@k adeptly discerns the varied characteristics of policy portfolios and identifies a safe and efficient estimator that adjusts to the specific budget ($k$) and problem instance ($J(\pi_b)$). Overall, the benchmark results suggest that SharpeRatio@k provides a more pragmatically meaningful comparison of OPE estimators than existing accuracy metrics.

***SharpeRatio@k and benchmark results suggest future research directions.*** The benchmarking results with SharpeRatio@k provide the following directions and suggestions for future OPE research.

1. **Future research should perform evaluation-of-OPE based on SharpeRatio@k:** The findings from the previous section suggest that SharpeRatio@k provides more actionable insights compared to traditional accuracy metrics. The benchmark results using SharpeRatio@k sometimes significantly differ from those obtained with conventional accuracy metrics. This highlights the importance of integrating SharpeRatio@k into future research to provide more insightful evaluation-of-OPE regarding the risk-return tradeoff and efficiency.

2. **A new estimator that explicitly optimizes the risk-return tradeoff:** While DR and MDR are generally regarded as advanced in existing literature, they do not consistently outperform DM, PDIS, and MIS according to SharpeRatio@k. This is attributable to their lack of specific design for optimizing the risk-return tradeoff and efficiency. Consequently, a promising research avenue would be to create a new estimator that explicitly focuses more on optimizing this risk-return tradeoff than existing methods.

3. **A new estimator selection method:** The results show that the most *efficient* estimator varies significantly across different environments, underscoring the need for adaptively selecting the most suitable estimator for reliable OPE. Given that existing estimator selection methods predominantly focus on "accuracy" metrics like MSE and Regret (Lee et al., 2022; Su et al., 2020; Udagawa et al., 2023; Xie & Jiang, 2021; Zhang & Jiang, 2021), there is an intriguing opportunity for future research to develop a novel estimator selection method that considers risks and efficiency.

## 5 RELATED WORK

**Evaluation-of-OPE Metrics**   The pursuit of accurately evaluating counterfactual policies has traditionally been a cornerstone of OPE research (Uehara et al., 2022; Saito et al., 2021b). Consequently, the Mean-Squared Error (MSE) has become the go-to "accuracy" metric, with the comprehensive study by Voloshin et al. (2019) exploring how the MSE of OPE estimators varies with trajectory lengths and degrees of distribution shift. In a different approach, Doroudi et al. (2017) highlights instances of biased policy selection caused by OPE and advocates for evaluating estimators based on their capability to identify the most effective policy, aligning with the concept of the regret metric. Subsequent benchmarks (Fu et al., 2021; Tang & Wiens, 2021) have compared estimators on their regret and rank correlation in downstream policy selection, but these metrics remain focused solely on the "accuracy" of policy selection, neglecting the risk-return tradeoff in subsequent A/B tests.

In contrast, SharpeRatio@k brings a novel perspective by emphasizing the risk-return tradeoff in OPE, drawing upon principles from portfolio management (Sharpe, 1998). We demonstrate that SharpeRatio@k can identify the most efficient estimator under various behavior policies, offering valuable insights for more effective estimator evaluation and selection in both academic and practical settings. Our research aligns somewhat with the findings of Kurenkov & Kolesnikov (2022), who show that preferred offline RL algorithms vary depending on the number of hyperparameters tested online. However, their work only assesses the expected performance of the best policy in the top-$k$ deployment, overlooking potential risks encountered during online testing. Additionally, their focus is on comparing offline RL methods, whereas our study centers on the evaluation of OPE.

**Risk-Return Tradeoff Metrics in Statistics and Finance**   Risk assessment plays a pivotal role in high-stakes fields such as finance, autonomous driving, and healthcare, ensuring safety and informed decision-making. Commonly used risk assessment metrics include quartile measures like the $\alpha$-quartile range, Value at Risk (VaR), and Conditional VaR (CVaR). For instance, CVaR calculates the average outcome in the worst $\alpha\%$ of cases. These metrics are recognized for effectively gauging the risk in decisions based on their worst-case scenarios (Rockafellar et al., 2000), yet they fall short in evaluating the risk-return tradeoff. The Sharpe Ratio, a concept widely applied in various contexts, particularly addresses this by evaluating the return of a decision relative to its associated risk. Originally conceptualized in finance (Sharpe, 1998), it calculates the return on investment (the asset's end-period price minus its purchase price) adjusted for the asset price's volatility (standard deviation during the period), thereby facilitating discussions on investment efficiency.

Our work draws inspiration from the Sharpe Ratio as defined in portfolio management (Sharpe, 1998) and adapts it for the first time as a metric to assess OPE estimators. The core idea is to view the top-$k$ policies chosen by an OPE estimator as a "policy portfolio". In this context, SharpeRatio@k measures the "efficiency" of this policy portfolio by balancing the "return" (performance improvement) against the "risk" of implementing suboptimal policies during the A/B testing phase. Our experimental results demonstrate that this application of SharpeRatio@k provides a more insightful and relevant evaluation of OPE estimators compared to conventional accuracy metrics.

## 6 CONCLUSION

This work proposed the concept of "efficiency" in Off-Policy Evaluation (OPE) and introduced a novel evaluation metric, SharpeRatio@k. Illustrative examples revealed that while existing metrics focus solely on the "accuracy" of OPE and Off-Policy Selection (OPS), they fail to capture the crucial differences in the risk-return tradeoff of various OPE estimators. We further show that SharpeRatio@k effectively assesses the efficiency of OPE estimators, taking into account specific problem instances, such as the performance of the behavior policy and online evaluation budget. The insights gained from our benchmark studies, using both SharpeRatio@k and SCOPE-RL, offer valuable guidance and recommendations for future research in OPE.

### ACKNOWLEDGMENTS

We would like to thank Daniel Cao and Romain Deffayet for their valuable feedback on the manuscript. We would also like to thank anonymous reviewers for their helpful feedback.

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

# A  EXPERIMENT DETAILS AND ADDITIONAL RESULTS

## A.1  DEFINITION OF COMPARED ESTIMATORS

**Direct Method (DM)**   DM is a model-based approach, which uses the initial state value estimated by Fitted Q Evaluation (FQE) (Le et al., 2019).[4] It first learns the Q-function from the logged data via temporal-difference (TD) learning and then utilizes the estimated Q-function for OPE as follows.

$$\hat{J}_{\mathrm{DM}}(\pi; \mathcal{D}) := \frac{1}{n} \sum_{i=1}^{n} \sum_{a \in \mathcal{A}} \pi(a|s_0^{(i)}) \hat{Q}(s_0^{(i)}, a),$$

where $\hat{Q}(s_t, a_t)$ is an estimated state-action value. DM has lower variance compared to other estimators, but often produces large bias due to approximation errors of the Q-function (Jiang & Li, 2016; Thomas & Brunskill, 2016).

**(Self-Normalized) Per-Decision Importance Sampling ((SN)PDIS)**   PDIS applies importance sampling to correct the distribution shift between $\pi_b$ and $\pi$. PDIS also leverages the sequential nature of the MDP to reduce the variance of trajectory-wise importance sampling. Specifically, since $s_t$ only depends on the states and actions observed previously (i.e., $s_0, \ldots, s_{t-1}$ and $a_0, \ldots, a_{t-1}$) and is independent of those observed in future time steps (i.e., $s_{t+1}, \ldots, s_T$ and $a_{t+1}, \ldots, a_T$), PDIS considers only the importance weights related to past interactions for each time step as follows (Precup et al., 2000).

$$\hat{J}_{\mathrm{PDIS}}(\pi; \mathcal{D}) := \frac{1}{n} \sum_{i=1}^{n} \sum_{t=0}^{T-1} \gamma^t w_{0:t}^{(i)} r_t^{(i)},$$

where $w_{0:t} := \prod_{t'=0}^{t} (\pi(a_{t'} \mid s_{t'})/\pi_b(a_{t'} \mid s_{t'}))$ represents the importance weight with respect to the previous action choices for time step $t$. PDIS is unbiased but often introduces substantial variance (Jiang & Li, 2016; Thomas & Brunskill, 2016). To alleviate the variance issue, we use the following *self-normalized* version (Kallus & Uehara, 2019) of PDIS in our experiments.

$$\hat{J}_{\mathrm{SNPDIS}}(\pi; \mathcal{D}) := \sum_{i=1}^{n} \sum_{t=0}^{T-1} \gamma^t \frac{w_{0:t}^{(i)}}{\sum_{i'=1}^{n} w_{0:t}^{(i')}} r_t^{(i)},$$

where $w_{0:t}/(\sum_{i=1}^{n} w_{0:t})$ is the self-normalized importance weight. The self-normalized estimator is no longer unbiased, but it retains the consistency of PDIS. Despite their reduced variance compared to trajectory-wise variants, (SN)PDIS might still suffer from high variance when $T$ is large.

**(Self-Normalized) Doubly Robust ((SN)DR)**   DR is a hybrid of model-based estimation and importance sampling (Dudík et al., 2011). It introduces $\hat{Q}$ as a baseline estimation in the recursive form of PDIS and applies importance weighting only to its residual (Jiang & Li, 2016; Thomas & Brunskill, 2016).

$$\hat{J}_{\mathrm{DR}}(\pi; \mathcal{D}) := \frac{1}{n} \sum_{i=1}^{n} \sum_{t=0}^{T-1} \gamma^t \left( w_{0:t}^{(i)}(r_t^{(i)} - \hat{Q}(s_t^{(i)}, a_t^{(i)})) + w_{0:t-1}^{(i)} \sum_{a \in \mathcal{A}} \pi(a|s_t^{(i)}) \hat{Q}(s_t^{(i)}, a) \right),$$

where $\hat{Q}$ works as a control variate. DR is unbiased and reduces the variance of PDIS when $\hat{Q}(\cdot)$ is reasonably accurate. However, it can still have high variance when the trajectory length $T$ is large (Fu et al., 2021) or the action space $|\mathcal{A}|$ is large (Saito & Joachims, 2022; Saito et al., 2023) is large. Again, we use the following self-normalized version of DR in our experiment.

$$\hat{J}_{\mathrm{SNDR}}(\pi; \mathcal{D})$$
$$:= \sum_{i=1}^{n} \sum_{t=0}^{T-1} \gamma^t \left( \frac{w_{0:t}^{(i)}}{\sum_{i'=1}^{n} w_{0:t}^{(i')}} (r_t^{(i)} - \hat{Q}(s_t^{(i)}, a_t^{(i)})) + \frac{w_{0:t-1}^{(i)}}{\sum_{i'=1}^{n} w_{0:t-1}^{(i')}} \sum_{a \in \mathcal{A}} \pi(a|s_t^{(i)}) \hat{Q}(s_t^{(i)}, a) \right).$$

---

[4]Our open source software, SCOPE-RL, uses the implementation of FQE provided by d3rlpy (Seno & Imai, 2021).

**Marginalized Importance Sampling estimators**   When the trajectory length ($T$) is large, the variance of PDIS and DR can be very high. This issue is often referred to as "the curse of horizon" in OPE (Liu et al., 2018). To alleviate this variance issue of estimators that rely on importance sampling with respect to the policies, several estimators utilize state marginal or state-action marginal importance weights, which are defined as follows (Liu et al., 2018; Uehara et al., 2020):

$$\rho(s, a) := d^\pi(s, a)/d^{\pi_b}(s, a), \qquad\qquad \rho(s) := d^\pi(s)/d^{\pi_b}(s)$$

where $d^\pi(s, a)$ and $d^\pi(s)$ are the marginal visitation probability of the policy $\pi$ on $(s, a)$ or $s$, respectively. We use the state-action marginal importance weight in our experiment, resulting in the following Marginal Importance Sampling (MIS) and Marginal Doubly Robust (MDR) estimators.

$$\hat{J}_{\text{SNMIS}}(\pi; \mathcal{D}) := \sum_{i=1}^{n} \sum_{t=0}^{T-1} \gamma^t \frac{\rho(s_t^{(i)}, a_t^{(i)})}{\sum_{i'=1}^{n} \rho(s_t^{(i')}, a_t^{(i')})} r_t^{(i)},$$

$$\hat{J}_{\text{SNMDR}}(\pi; \mathcal{D})$$
$$:= \frac{1}{n} \sum_{i=1}^{n} \sum_{a \in \mathcal{A}} \pi(a|s_0^{(i)}) \hat{Q}(s_0^{(i)}, a)$$
$$+ \sum_{i=1}^{n} \sum_{t=0}^{T-1} \gamma^t \frac{\rho(s_t^{(i)}, a_t^{(i)})}{\sum_{i'=1}^{n} \rho(s_t^{(i')}, a_t^{(i')})} \left( r_t^{(i)} + \gamma \sum_{a \in \mathcal{A}} \pi(a|s_t^{(i)}) \hat{Q}(s_{t+1}^{(i)}, a) - \hat{Q}(s_t^{(i)}, a_t^{(i)}) \right),$$

Note that, as given in the above definitions, we use self-normalized variants of these estimators in our experiment.

**Extension to the continuous action space**   When the action space is continuous, the original version of importance weight $w_t = \pi(a_t|s_t)/\pi_b(a_t|s_t) = (\pi(a|s_t)/\pi_b(a_t|s_t))\mathbb{I}(a = a_t)$ ends up rejecting all actions, as $\mathbb{I}(a = a_t)$ filters only the action observed in the logged data. To address this issue, continuous OPE estimators (PDIS and DR) apply the kernel density estimation technique to smooth the rejection sampling part as follows (Kallus & Zhou, 2018).

$$\overline{w}_t = \int_{a \in \mathcal{A}} \frac{\pi(a|s_t)}{\pi_b(a_t|s_t)} \cdot \frac{1}{h} K\left(\frac{a - a_t}{h}\right) da,$$

where $K(\cdot)$ denotes a kernel function and we use the Gaussian kernel $K(x) = \frac{1}{\sqrt{2\pi}} \exp(-x^2/2)$ in our experiments. $h$ is the bandwidth hyperparameter; a large value of $h$ leads to a high-bias but low-variance estimator, while a small value of $h$ results in a low-bias but high-variance estimator.

## A.2   OMITTED DETAILS OF THE BENCHMARK EXPERIMENTS

**Experiment setting for discrete control tasks**   The experimental procedures of discrete control tasks are similar to those of continuous control tasks. First, we train DDQN (Van Hasselt et al., 2016) in an online environment to learn a base Q-function $\hat{q}_b(s_t, a_t) \approx \mathbb{E}[\sum_{t'=t+1}^{T} \gamma^{t'-t} r_{t'}|s_t, a_t]$. Then, to obtain a stochastic behavior policy $\pi_b$, we apply the softmax function to the estimated Q-function as $\pi_b(a_t|s_t) := \frac{\exp(\hat{q}_b(s_t, a_t)/\tau)}{\sum_{a' \in \mathcal{A}} \exp(\hat{q}_b(s_t, a')/\tau)}$, where $\tau$ controls the stochasticity of $\pi_b$ (i.e., a large absolute value of $\tau$ leads to a near-uniform behavior policy, while a small absolute value of $\tau$ leads to a near-deterministic one). Next, we use the behavior policy $\pi_b$ to generate a logged dataset $\mathcal{D}^{(tr)}$, which is used to train a set of candidate policies ($\Pi$). To define candidate policies, we train CQL (Kumar et al., 2020) and BCQ (Fujimoto et al., 2019) with 3 different neural networks to estimate the Q-function $\hat{q}(s_t, a_t)$. Then, we define epsilon-greedy style policies as $\pi(a_t|s_t) := (1 - \epsilon)\mathbb{I}\{a_t = a^*\} + \epsilon/|\mathcal{A}|$, where $a^* := \max_{a' \in \mathcal{A}} \hat{q}(s_t, a')$ and $\epsilon$ controls the degree of exploration. We use 4 different values of $\epsilon$ for each of the base Q-functions, thus the total number of candidate policies is 24. From this assortment, we randomly select 10 policies to form candidate policies ($\Pi$) in each experiment.

**Baseline accuracy metrics**   In our experiments, we report the value of MSE and Regret normalized by the best performance among the candidate policies in $\Pi$. More formally, the following defines the

Table 2: Statistics of the environments used in the experiments

| env. | state dim. | action type | # of actions / action dim. | max episode steps |
|---|---|---|---|---|
| Reacher | 11 | continuous | 2 | 30 (50) |
| InvertedPendulum | 4 | continuous | 1 | 100 (1000) |
| Hopper | 11 | continuous | 3 | 30 (1000) |
| Swimmer | 8 | continuous | 2 | 100 (1000) |
| CartPole | 4 | discrete | 2 | 200 (500) |
| MountainCar | 2 | discrete | 3 | 200 (200) |
| Acrobot | 6 | discrete | 3 | 500 (500) |

*Note*: In the column "max episode steps", the values in parenthesis show the default value set by gym (Brockman et al., 2016), which means that we use smaller maximum episode steps than default. This is to avoid extremely large importance weights of PDIS. We also use the latest version of the environments provided by gym==0.26.2.

*normalized* version of MSE and Regret as used in our experiment.

$$\mathrm{nMSE}(\hat{J}) := \frac{\sum_{\pi \in \Pi}(\hat{J}(\pi; \mathcal{D}) - J(\pi))^2}{|\Pi| \cdot \max\{(\max_{\pi \in \Pi} J(\pi))^2, (\max_{\pi \in \Pi} J(\pi) - \min_{\pi \in \Pi} J(\pi))^2\}},$$

$$\mathrm{nRegret}@k(\hat{J}) := \frac{\max_{\pi \in \Pi} J(\pi) - \max_{\pi \in \Pi_k(\hat{J})} J(\pi)}{\max\{\max_{\pi \in \Pi} J(\pi), \max_{\pi \in \Pi} J(\pi) - \min_{\pi \in \Pi} J(\pi)\}}.$$

Note that $\max_{\pi \in \Pi} J(\pi)$ is used when the policy performance is always positive, while $\max_{\pi \in \Pi} J(\pi) - \min_{\pi \in \Pi} J(\pi)$ is used when the policy performance can be a negative value.

**Environments** The following describes the state, action, and reward settings of the environments used in our experiments. Table 2 summarizes the key statistics of each environment.

***Gym-Mujoco*** provides several continuous control tasks that simulate the physical movements of animals or objects. Among them, ***Reacher*** is a task to control two-jointed robot arms. Provided with a target position, the goal of Reacher is to move the fingertip of the robot arm close to the target. Thus, the action is 2-dimensional, corresponding to the torque applied to the joints of the arms, while the reward is the distance between the target and the fingertip. There are also penalties applied to the reward for taking large actions. The state observation consists of 11-dimensional features, including the positions and velocities of the two arms. ***Swimmer*** is a similar task, but its goal is to push a two-joined robot forward. Thus, the reward is defined by the difference in the robot's current and previous positions, which is penalized by the norm of actions. It also observes 8-dimensional states and 2-dimensional actions corresponding to the torque applied to joints. ***Hopper*** is another task that involves moving the hopper object forward; thus, the reward system is similar to that of Swimmer. Its state is 11-dimensional, and its action is 3-dimensional. Finally, ***InvertedPendulum*** is a task that involves controlling a cart in order to keep the pendulum on it upright. The state is 4-dimensional, corresponding to the position, velocity, and angle of the cart and pendulum, and the action is the force applied to the cart, which is 1-dimensional. When the pendulum is upright, a positive reward (+1) is given. If the pendulum falls, the reward becomes zero thereafter.

***Gym-Classic Control*** provides several discrete and continuous control tasks. ***CartPole*** is a task that involves controlling a cart to keep the pole on it upright, similar to InvertedPendulum. While the state and reward settings of CartPole are the same as those of InvertedPendulum, the action space for CartPole is discrete. Specifically, CartPole has only two actions, which correspond to pushing the cart right or left. ***MountainCar*** is a task where the goal is to control a car in order to climb a mountain. The agent receives a reward of -1 until the car reaches the top of the mountain. The state is 2-dimensional, corresponding to the position and velocity of the car, while there are 3 discrete actions: accelerating to the left, accelerating to the right, or no acceleration. Finally, ***Acrobot*** is a task that involves swinging a two-jointed chain upwards. The reward is -1 until the chain is swung up, and 3 discrete actions correspond to torques of +1, -1, or 0 applied to the chain. The states are 6-dimensional, corresponding to the position and velocity of the two chains.

**Difference between our benchmark design and those of (Fu et al., 2021; Voloshin et al., 2019)**
Previously, DOPE (Fu et al., 2021) and COBS (Voloshin et al., 2019) empirically evaluated OPE

estimators in continuous control tasks, including Gym-Mujoco, and in a discrete control task (MountainCar), respectively. While we share the choice of environments with these existing benchmarks, our experiment design differs from theirs, as we emphasize the evaluation of risks, returns, and efficiency of OPE estimators beyond their accuracy.

The most significant difference between DOPE (Fu et al., 2021) and our work is our use of more diverse and stochastic candidate policies. Specifically, while DOPE obtains evaluation policies from different training phases of a single SAC (Haarnoja et al., 2018) policy without adding explicit noise to actions, we first train two different offline RL algorithms with three different parameters and then add some stochastic noise to the action with four different noise levels. By doing this, we can obtain a candidate set of policies with varying policy values and create a more complex and diverse distributional shift between behavior and evaluation policies. Another difference is the derivation of importance weight. While we use the true (self-normalized) importance weight to define PDIS and DR, DOPE estimates the behavior policy $\pi_b(a_t|s_t)$ from the logged data. This results in more accurate estimation for PDIS and DR in our experiments compared to those in the DOPE experiments.[5] It is worth noting that precise behavior policy estimations are often accessible in applications such as e-commerce or music streaming, as behavior policy probabilities can easily be logged in these systems (Saito et al., 2021a; Kiyohara et al., 2021; 2022). Although COBS (Voloshin et al., 2019) also uses precise importance weights to define PDIS and DR, these estimators provide less accurate estimation than DM because COBS uses smaller datasets, e.g., 256, 512, and 1024 $(s, a, s', r)$-tuples, while ours and DOPE use more than 100K tuples.[6] Finally, while COBS investigates how the MSE of each estimator changes with various configurations, including the degree of distributional shift, data sizes, and the length of trajectories, COBS does not study policy selection as we and DOPE do.

**Models and parameters**  While our experimental code is publicly available[7], we also describe the implementation details of our experiments. Tables 3 and 4 summarize the models and parameters used for learning behavior policies, and Tables 5-11 describe those for learning candidate (evaluation) policies. Figures 28-34 also show the (on-policy) policy performance of the behavior and candidate policies used in our benchmark experiments. Lastly, Table 12 reports the bandwidth hyperparameter (of continuous OPE) and the networks and parameters of FQE and DICE, which are used for estimating the Q-function and marginal importance weights. Please note that we did not conduct extensive hyperparameter tuning for FQE and DICE in line with previous works (Fu et al., 2020; Qin et al., 2021; Voloshin et al., 2019) because hyperparameter tuning of OPE estimators using only logged data is a non-trivial task and is of independent interest.

**Computation**  We conducted benchmark experiments for discrete control tasks on an M1 MacBook Pro. The computational time greatly varied among benchmark environments, taking about 3 hours for CartPole, 4 hours for MountainCar, and 8 hours for Acrobot. For experiments on continuous control tasks, we used Google Colaboratory utilizing the V100 GPU. These experiments took about 6 hours for Reacher, 10 hours for InvertedPendulum, 6 hours for Hopper, and 10 hours for Swimmer.

## A.3 ADDITIONAL RESULTS AND DISCUSSIONS

Figures 7-13 illustrate SharpeRatio@k and its decomposition into returns over a risk-free baseline (i.e., numerator) and risks (i.e., denominator) for varying online evaluation budgets $k$. As references, Figures 14-20 show the statistics of the top-$k$ policy portfolio formed by each OPE estimator. Figures 21-27 compare the results using conventional metrics, SharpeRatio@4, and SharpeRatio@7.

***Are low-variance estimators always evaluated as safe and efficient by SharpeRatio@k?***  In the MountainCar example from Section 4, we noted that a low-variance estimator (MDR or DM) was significantly more efficient than the most "accurate" (in terms of MSE) and the highest-return (Regret) but high-variance estimator, i.e., MIS. Although SharpeRatio@k has been shown to yield reasonable results in the main text, one might wonder: *Does SharpeRatio@k invariably favor a low-variance estimator over a high-variance one?* Our benchmark results definitively indicate that this is not the case, with the Hopper environment serving as a counterexample. In Figure 9, we

---

[5]We use a small "max episode steps" setting to avoid extreme importance weights that may produce "NaNs".

[6]The number of data tuples is calculated as "number of trajectories" $\times$ "max episode steps" in our benchmark.

[7]https://github.com/hakuhodo-technologies/scope-rl/tree/main/experiments

observe that the standard deviation at $k$ (std@$k$) for DM and MIS are among the lowest compared to other estimators and are nearly identical. Yet, SharpeRatio@k assesses these two estimators quite differently – MIS ranks as the best among the compared estimators, whereas DM ranks as the worst. This discrepancy arises because MIS consistently selects well-performing policies, while DM tends to overestimate poor-performing policies, as depicted in Figure 18. SharpeRatio@k effectively discerns this critical difference by considering both returns (best@$k$ in the numerator) and risks (std@$k$ in the denominator). These findings underscore that the trade-off between risks and returns is crucial, and the denominator (std@$k$) alone is insufficient for determining a safe and efficient OPE estimator.

***How does evaluation-of-OPE results on SharpeRatio@k change with varying online evaluation budgets ($k$)?*** Next, we examine how SharpeRatio@$k$ varies among small values of $k$ (e.g., $k = 2, 3$), medium values ($k = 4, 5, 6$), and large values ($k \geq 7$) as illustrated in Figures 21-27. Overall, we notice that SharpeRatio@$k$ delivers consistent evaluations for $k \geq 4$ in most benchmark environments, except for CartPole and MountainCar. This consistency is primarily because the policy portfolio chosen by each estimator tends to converge to a similar portfolio as $k$ increases, resulting in less divergence in the values of SharpeRatio@$k$ among OPE estimators. As highlighted in the main text, MountainCar (Figure 13) is an outlier, indicating that an efficient OPE estimator can sometimes vary with the online evaluation budget ($k$). Furthermore, we observe that SharpeRatio@$k$ for $k \leq 3$ can drastically differ from those for $k \geq 4$ due to significant fluctuations in std@$k$. This suggests that top-$k$ policy selection often involves considerable uncertainty when $k \leq 3$. Consequently, we would recommend implementing at least 4 policies in online A/B tests as a prudent strategy to achieve stable policy selection results and reliable evaluations of OPE estimator efficiency.

***How should we interpret the results when SharpeRatio and other metrics behave differently?*** As highlighted in the main text, the estimator selection results under SharpeRatio@$k$ sometimes deviate from those determined by other metrics. This divergence occurs because SharpeRatio considers risk, while other metrics overlook this factor. Specifically, current metrics primarily focus on the accuracy of OPE (MSE), the alignment of resulting policies (RankCorr), or the top-1 policy selection (Regret). In contrast, SharpeRatio evaluates the enhancement in the performance of the production policy, discounting it by the risk incurred during online A/B tests. Ideally, an estimator would excel in all metrics. However, when an estimator is favored by conventional metrics but assessed differently under SharpeRatio, it is indicative of the estimator being high-return yet high-risk. Consequently, in critical scenarios, such as in medical fields, SharpeRatio should be given precedence. This is because SharpeRatio is the sole metric that acknowledges safety and can adeptly measure the risk-return tradeoff in the two-stage policy selection process, which involves online A/B tests.

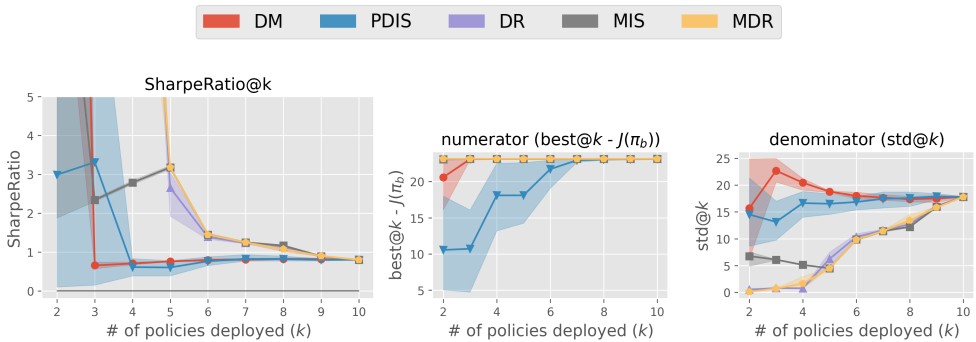

Figure 7: Estimators' performance comparison with **SharpeRatio@k** in **Reacher**. A higher value is better for SharpeRatio@k and the numerator, while a lower value is better for the denominator.

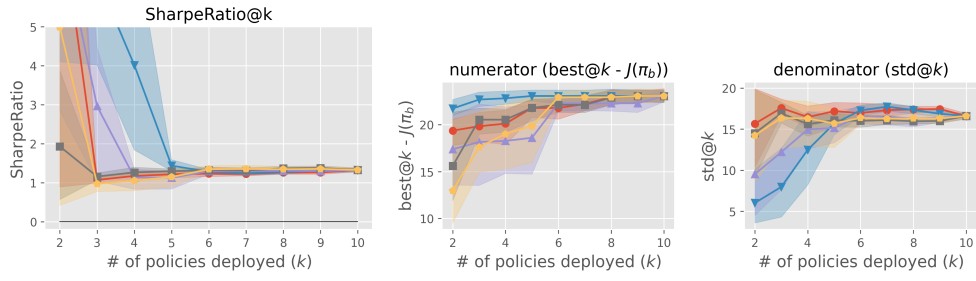

Figure 8: Estimators' performance comparison with **SharpeRatio@k** in **InvertedPendulum**. A higher value is better for SharpeRatio@k and the numerator, while a lower value is better for the denominator.

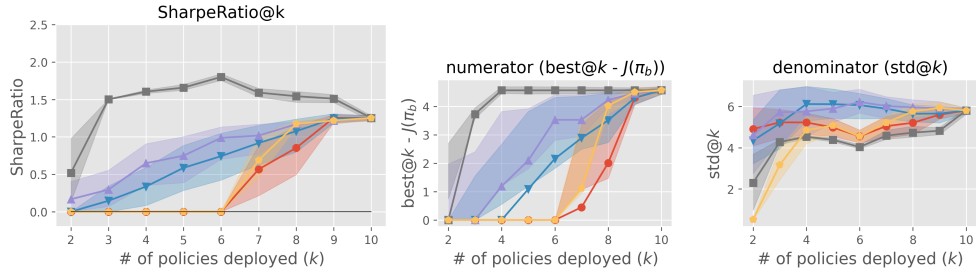

Figure 9: Estimators' performance comparison with **SharpeRatio@k** in **Hopper**. A higher value is better for SharpeRatio@k and the numerator, while a lower value is better for the denominator.

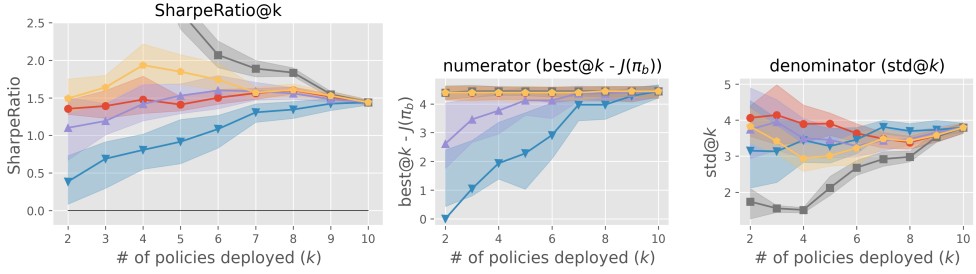

Figure 10: Estimators' performance comparison with **SharpeRatio@k** in **Swimmer**. A higher value is better for SharpeRatio@k and the numerator, while a lower value is better for the denominator.

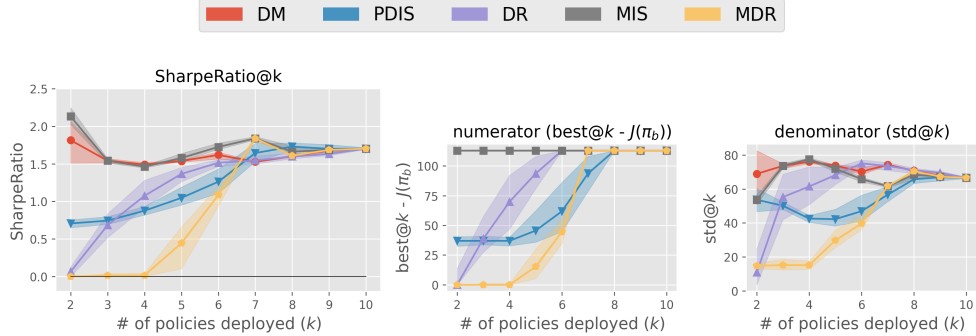

Figure 11: Estimators' performance comparison with **SharpeRatio@k** in **CartPole**. A higher value is better for SharpeRatio@k and the numerator, while a lower value is better for the denominator.

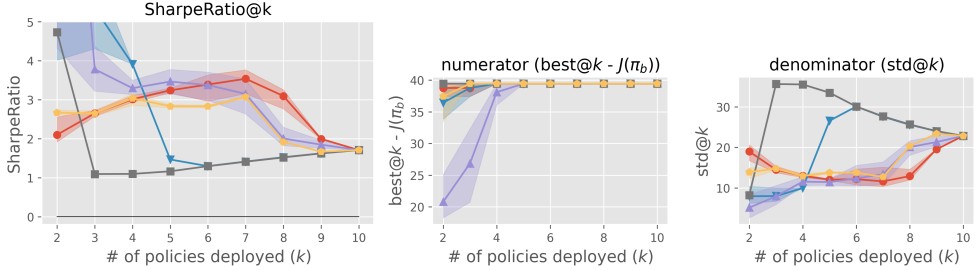

Figure 12: Estimators' performance comparison with **SharpeRatio@k** in **MountainCar**. A higher value is better for SharpeRatio@k and the numerator, while a lower value is better for the denominator.

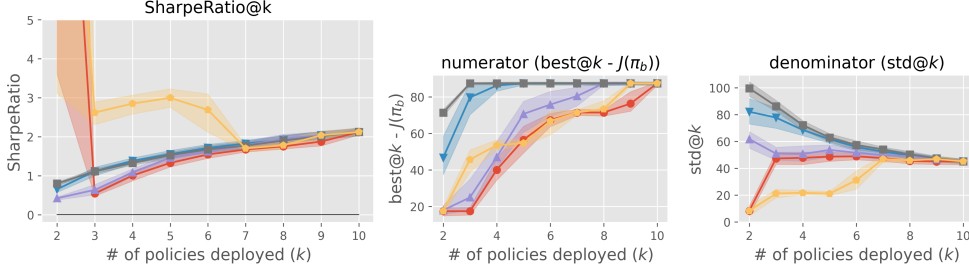

Figure 13: Estimators' performance comparison with **SharpeRatio@k** in **Acrobot**. A higher value is better for SharpeRatio@k and the numerator, while a lower value is better for the denominator.

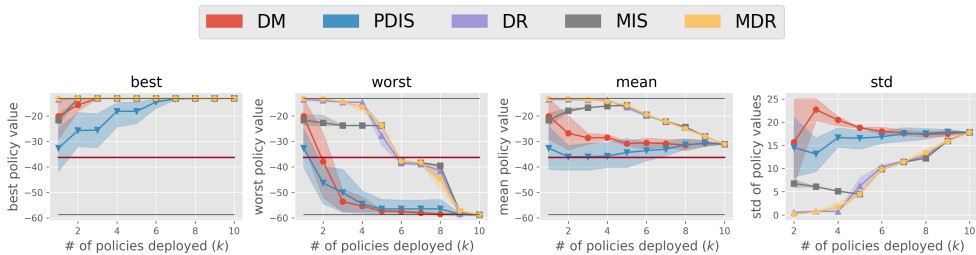

Figure 14: **Statistics of the top-$k$ policy portfolio** of each estimator in **Reacher**. A lower value is better for std, while a higher value is better for best, worst, and mean policy performances. The dark red line shows the performance of $\pi_b$, which is the risk-free baseline of SharpeRatio@k.

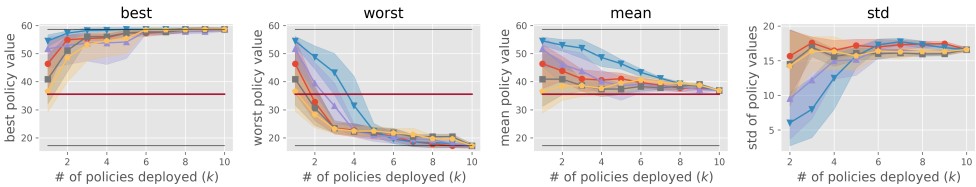

Figure 15: **Statistics of the top-$k$ policy portfolio** of each estimator in **InvertedPendulum**. A lower value is better for std, while a higher value is better for best, worst, and mean policy performances. The dark red line shows the performance of $\pi_b$, which is the risk-free baseline of SharpeRatio@k.

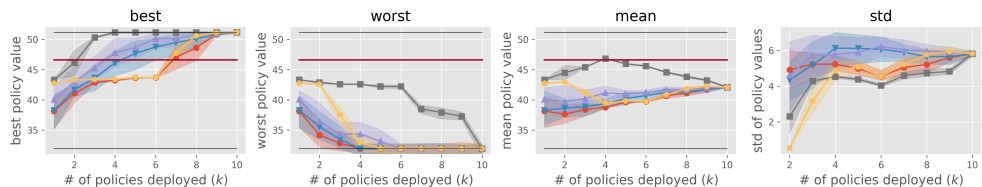

Figure 16: **Statistics of the top-$k$ policy portfolio** of each estimator in **Hopper**. A lower value is better for std, while a higher value is better for best, worst, and mean policy performances. The dark red line shows the performance of $\pi_b$, which is the risk-free baseline of SharpeRatio@k.

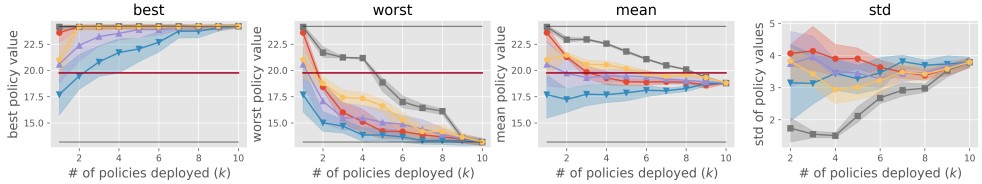

Figure 17: **Statistics of the top-$k$ policy portfolio** of each estimator in **Swimmer**. A lower value is better for std, while a higher value is better for best, worst, and mean policy performances. The dark red line shows the performance of $\pi_b$, which is the risk-free baseline of SharpeRatio@k.

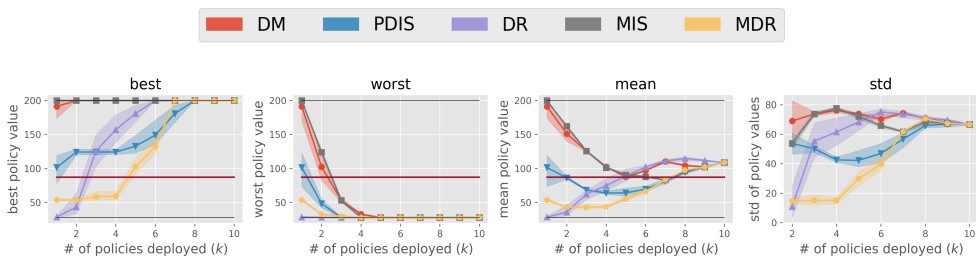

Figure 18: **Statistics of the top-$k$ policy portfolio** of each estimator in **CartPole**. A lower value is better for std, while a higher value is better for best, worst, and mean policy performances. The dark red line shows the performance of $\pi_b$, which is the risk-free baseline of SharpeRatio@k.

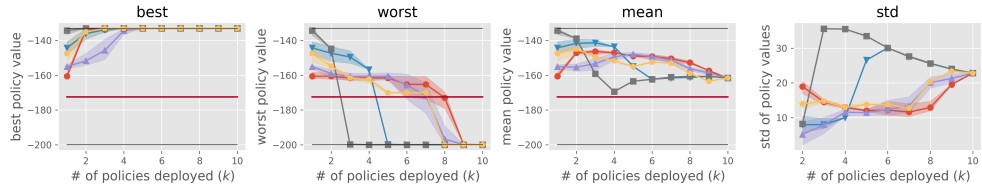

Figure 19: **Statistics of the top-$k$ policy portfolio** of each estimator in **MountainCar**. A lower value is better for std, while a higher value is better for best, worst, and mean policy performances. The dark red line shows the performance of $\pi_b$, which is the risk-free baseline of SharpeRatio@k.

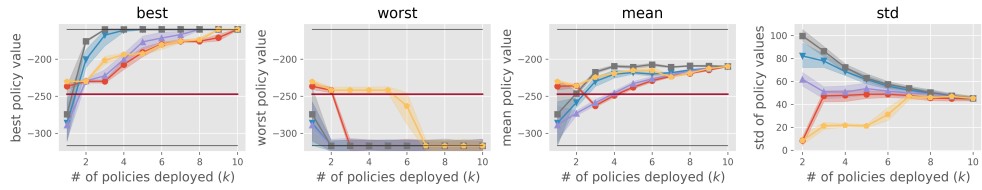

Figure 20: **Statistics of the top-$k$ policy portfolio** of each estimator in **Acrobot**. A lower value is better for std, while a higher value is better for best, worst, and mean policy performances. The dark red line shows the performance of $\pi_b$, which is the risk-free baseline of SharpeRatio@k.

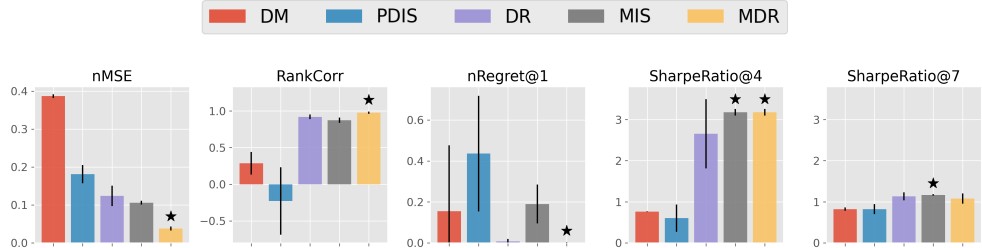

Figure 21: Estimators' performance comparison with **conventional metrics, SharpeRatio@4, and SharpeRatio@7** in **Reacher**. A lower value is better for nMSE and nRegret@1, while a higher value is better for RankCorr and SharpeRatio. The star (⋆) indicates the best estimator(s).

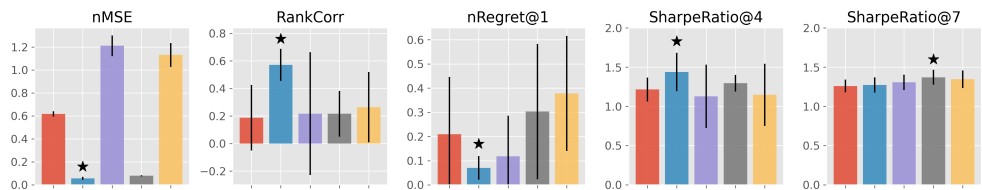

Figure 22: Estimators' performance comparison with **conventional metrics, SharpeRatio@4, and SharpeRatio@7** in **InvertedPendulum**. A lower value is better for nMSE and nRegret@1, while a higher value is better for RankCorr and SharpeRatio. The star (⋆) indicates the best estimator(s).

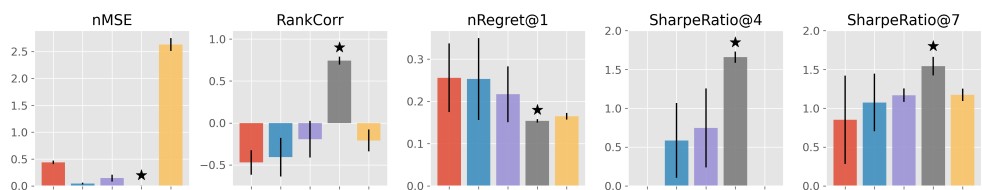

Figure 23: Estimators' performance comparison with **conventional metrics, SharpeRatio@4, and SharpeRatio@7** in **Hopper**. A lower value is better for nMSE and nRegret@1, while a higher value is better for RankCorr and SharpeRatio. The star (⋆) indicates the best estimator(s).

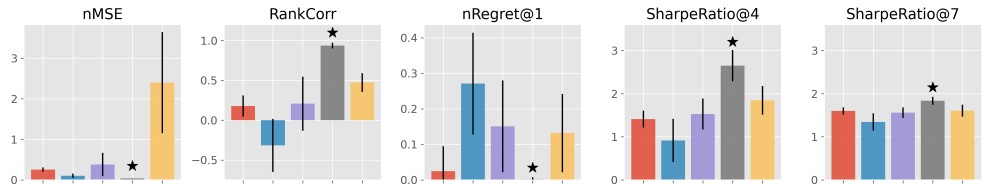

Figure 24: Estimators' performance comparison with **conventional metrics, SharpeRatio@4, and SharpeRatio@7** in **Swimmer**. A lower value is better for nMSE and nRegret@1, while a higher value is better for RankCorr and SharpeRatio. The star (⋆) indicates the best estimator(s).

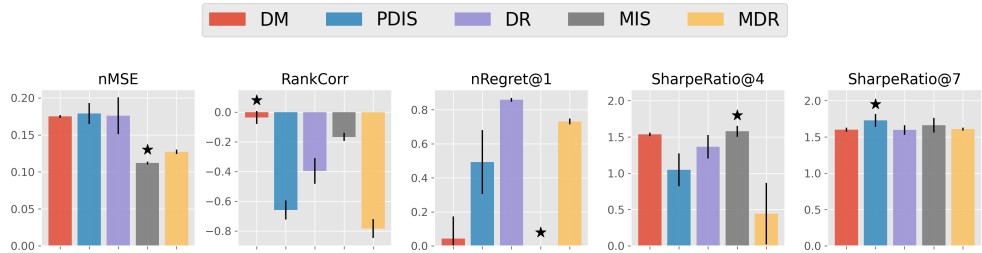

Figure 25: Estimators' performance comparison with **conventional metrics, SharpeRatio@4, and SharpeRatio@7** in **CartPole**. A lower value is better for nMSE and nRegret@1, while a higher value is better for RankCorr and SharpeRatio. The star (⋆) indicates the best estimator(s).

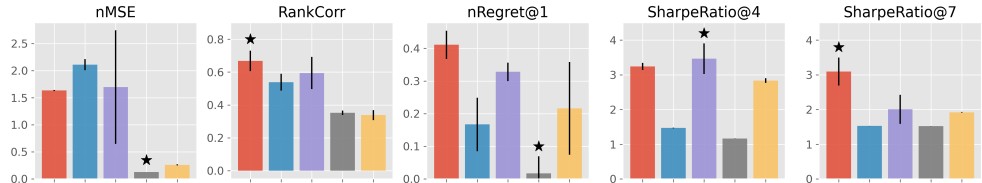

Figure 26: Estimators' performance comparison with **conventional metrics, SharpeRatio@4, and SharpeRatio@7** in **MountainCar**. A lower value is better for nMSE and nRegret@1, while a higher value is better for RankCorr and SharpeRatio. The star (⋆) indicates the best estimator(s).

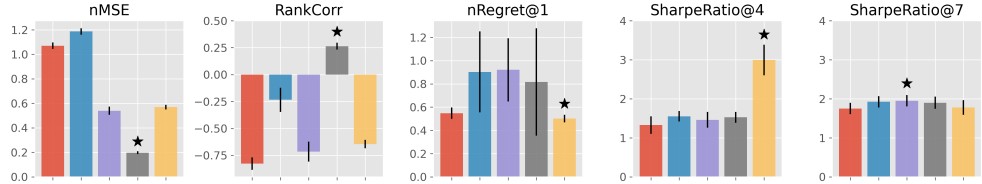

Figure 27: Estimators' performance comparison with **conventional metrics, SharpeRatio@4, and SharpeRatio@7** in **Acrobot**. A lower value is better for nMSE and nRegret@1, while a higher value is better for RankCorr and SharpeRatio. The star (⋆) indicates the best estimator(s).

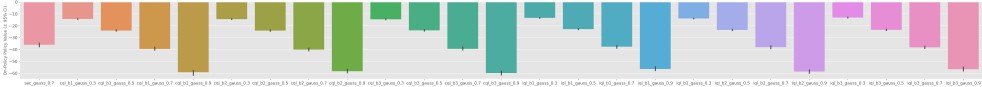

Figure 28: On-policy policy value of the behavior policy (leftmost) and the candidate policies (others) in Reacher

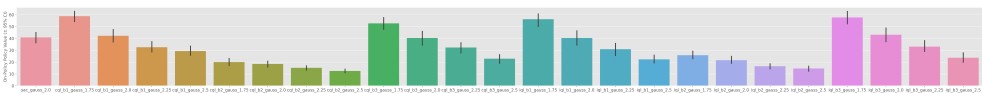

Figure 29: On-policy policy value of the behavior policy (leftmost) and the candidate policies (others) in InvertedPendulum

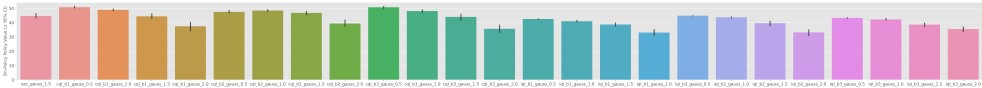

Figure 30: On-policy policy value of the behavior policy (leftmost) and the candidate policies (others) in Hopper

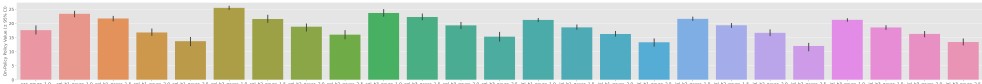

Figure 31: On-policy policy value of the behavior policy (leftmost) and the candidate policies (others) in Swimmer

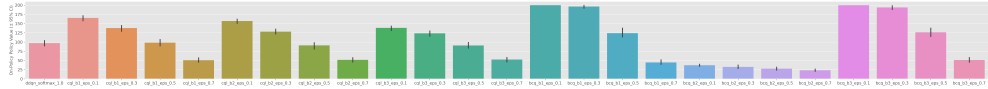

Figure 32: On-policy policy value of the behavior policy (leftmost) and the candidate policies (others) in CartPole

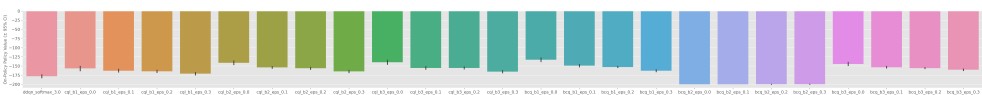

Figure 33: On-policy policy value of the behavior policy (leftmost) and the candidate policies (others) in MountainCar

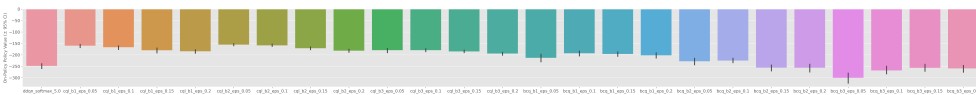

Figure 34: On-policy policy value of the behavior policy (leftmost) and the candidate policies (others) in Acrobot

Table 3: Models and parameters used to train the behavior policies in continuous control tasks

| env. | model | parameter | value |
|---|---|---|---|
| Reacher | SAC (Haarnoja et al., 2018) | actor_encoder_factory* | VectorEncoderFactory(hidden_units=[100]) |
| | | critic_encoder_factory* | VectorEncoderFactory(hidden_units=[100]) |
| | | q_func_factory* | MeanQFunctionFactory() |
| | | actor_learning_rate* | 1e-3 |
| | | critic_learning_rate* | 1e-3 |
| | | temp_learning_rate* | 1e-3 |
| | | batch_size* | 32 |
| | | n_steps* | 10000 |
| | | update_start_step* | 1000 |
| | | (randomization) class$^{\diamond\diamond}$ | GaussianHead |
| | | (randomization) sigma$^{\diamond}$ | 0.7 |
| InvertedPendulum | SAC (Haarnoja et al., 2018) | actor_encoder_factory* | VectorEncoderFactory(hidden_units=[100]) |
| | | critic_encoder_factory* | VectorEncoderFactory(hidden_units=[100]) |
| | | q_func_factory* | MeanQFunctionFactory() |
| | | actor_learning_rate* | 1e-3 |
| | | critic_learning_rate* | 1e-3 |
| | | temp_learning_rate* | 1e-3 |
| | | batch_size* | 32 |
| | | n_steps* | 10000 |
| | | update_start_step* | 1000 |
| | | (randomization) class$^{\diamond\diamond}$ | GaussianHead |
| | | (randomization) sigma$^{\diamond}$ | 2.0 |
| Hopper | SAC (Haarnoja et al., 2018) | actor_encoder_factory* | VectorEncoderFactory(hidden_units=[100]) |
| | | critic_encoder_factory* | VectorEncoderFactory(hidden_units=[100]) |
| | | q_func_factory* | MeanQFunctionFactory() |
| | | actor_learning_rate* | 1e-3 |
| | | critic_learning_rate* | 1e-3 |
| | | temp_learning_rate* | 1e-3 |
| | | batch_size* | 32 |
| | | n_steps* | 10000 |
| | | update_start_step* | 1000 |
| | | (randomization) class$^{\diamond\diamond}$ | GaussianHead |
| | | (randomization) sigma$^{\diamond}$ | 1.5 |
| Swimmer | SAC (Haarnoja et al., 2018) | actor_encoder_factory* | VectorEncoderFactory(hidden_units=[100]) |
| | | critic_encoder_factory* | VectorEncoderFactory(hidden_units=[100]) |
| | | q_func_factory* | MeanQFunctionFactory() |
| | | actor_learning_rate* | 1e-3 |
| | | critic_learning_rate* | 1e-3 |
| | | temp_learning_rate* | 1e-3 |
| | | batch_size* | 32 |
| | | n_steps* | 10000 |
| | | update_start_step* | 1000 |
| | | (randomization) class$^{\diamond\diamond}$ | GaussianHead |
| | | (randomization) sigma$^{\diamond}$ | 1.0 |

*Note*: We use the implementations provided by d3rlpy (Seno & Imai, 2021). $*$ and $**$ indicate the parameters and classes of d3rlpy, while $\diamond$ and $\diamond\diamond$ indicate those of our software. For the other parameters that are not specified in the table, we use the default values set by d3rlpy.

Table 4: Models and parameters used to train the behavior policies in discrete control tasks

| env. | model | parameter | value |
|------|-------|-----------|-------|
| CartPole | DDQN (Van Hasselt et al., 2016) | encoder_factory* | VectorEncoderFactory(hidden_units=[100]) |
| | | q_func_factory* | MeanQFunctionFactory() |
| | | target_update_interval* | 100 |
| | | batch_size* | 32 |
| | | learning_rate* | 1e-3 |
| | | n_steps* | 8000 |
| | | update_start_step* | 1000 |
| | | (replay buffer) maxlen* | 10000 |
| | | (explorer) class** | LinearDecayEpsilonGreedy |
| | | (explorer) start_epsilon* | 1.0 |
| | | (explorer) end_epsilon* | 0.01 |
| | | (explorer) duration* | 8000 |
| | | (randomization) class$^{\diamond\diamond}$ | SoftmaxHead |
| | | (randomization) tau$^{\diamond}$ | 1.0 |
| MountainCar | DDQN (Van Hasselt et al., 2016) | encoder_factory* | VectorEncoderFactory(hidden_units=[100]) |
| | | q_func_factory* | MeanQFunctionFactory() |
| | | target_update_interval* | 100 |
| | | batch_size* | 32 |
| | | learning_rate* | 1e-3 |
| | | n_steps* | 50000 |
| | | update_start_step* | 1000 |
| | | (replay buffer) maxlen* | 10000 |
| | | (explorer) class** | LinearDecayEpsilonGreedy |
| | | (explorer) start_epsilon* | 1.0 |
| | | (explorer) end_epsilon* | 0.01 |
| | | (explorer) duration* | 20000 |
| | | (randomization) class$^{\diamond\diamond}$ | SoftmaxHead |
| | | (randomization) tau$^{\diamond}$ | 3.0 |
| Acrobot | DDQN (Van Hasselt et al., 2016) | encoder_factory* | VectorEncoderFactory(hidden_units=[100]) |
| | | q_func_factory* | MeanQFunctionFactory() |
| | | target_update_interval* | 100 |
| | | target_update_interval* | 100 |
| | | batch_size* | 32 |
| | | learning_rate* | 5e-5 |
| | | n_steps* | 20000 |
| | | update_start_step* | 1000 |
| | | (replay buffer) maxlen* | 10000 |
| | | (explorer) class** | LinearDecayEpsilonGreedy |
| | | (explorer) start_epsilon* | 1.0 |
| | | (explorer) end_epsilon* | 0.1 |
| | | (explorer) duration* | 10000 |
| | | (randomization) class$^{\diamond\diamond}$ | SoftmaxHead |
| | | (randomization) tau$^{\diamond}$ | 5.0 |

*Note*: We use the implementations provided by d3rlpy (Seno & Imai, 2021). $*$ and $**$ indicate the parameters and classes of d3rlpy, while $\diamond$ and $\diamond\diamond$ indicate those of our software. For the other parameters that are not specified in the table, we use the default values set by d3rlpy.

Table 5: Models and parameters used to train the candidate policies in Reacher

| model | parameter | value |
|---|---|---|
| CQL (Kumar et al., 2020) | actor_encoder_factory* | VectorEncoderFactory(hidden_units=[100]) |
| | | VectorEncoderFactory(hidden_units=[30, 30]) |
| | | VectorEncoderFactory(hidden_units=[50, 10]) |
| | critic_encoder_factory* | VectorEncoderFactory(hidden_units=[100]) |
| | | VectorEncoderFactory(hidden_units=[30, 30]) |
| | | VectorEncoderFactory(hidden_units=[50, 10]) |
| | q_func_factory* | MeanQFunctionFactory() |
| | (randomization) class$^{\diamond\diamond}$ | GaussianHead |
| | (randomization) sigma$^\diamond$ | {0.3, 0.5, 0.7, 0.9} |
| IQL (Kostrikov et al., 2022) | actor_encoder_factory* | VectorEncoderFactory(hidden_units=[100]) |
| | | VectorEncoderFactory(hidden_units=[30, 30]) |
| | | VectorEncoderFactory(hidden_units=[50, 10]) |
| | critic_encoder_factory* | VectorEncoderFactory(hidden_units=[100]) |
| | | VectorEncoderFactory(hidden_units=[30, 30]) |
| | | VectorEncoderFactory(hidden_units=[50, 10]) |
| | (randomization) class$^{\diamond\diamond}$ | GaussianHead |
| | (randomization) sigma$^\diamond$ | {0.3, 0.5, 0.7, 0.9} |

*Note*: We use the implementations provided by d3rlpy (Seno & Imai, 2021). $*$ and $**$ indicate the parameters and classes of d3rlpy, while $\diamond$ and $\diamond\diamond$ indicate those of our software. For the other parameters that are not specified in the table, we use the default values set by d3rlpy. Note that since we use three different encoders and four different values of sigma, we obtain 12 different policies for both CQL and IQL.

Table 6: Models and parameters used to train the candidate policies in InvertedPendulum

| model | parameter | value |
|---|---|---|
| CQL (Kumar et al., 2020) | actor_encoder_factory* | VectorEncoderFactory(hidden_units=[100]) |
| | | VectorEncoderFactory(hidden_units=[30, 30]) |
| | | VectorEncoderFactory(hidden_units=[50, 10]) |
| | critic_encoder_factory* | VectorEncoderFactory(hidden_units=[100]) |
| | | VectorEncoderFactory(hidden_units=[30, 30]) |
| | | VectorEncoderFactory(hidden_units=[50, 10]) |
| | q_func_factory* | MeanQFunctionFactory() |
| | (randomization) class$^{\diamond\diamond}$ | GaussianHead |
| | (randomization) sigma$^\diamond$ | {1.75, 2.0, 2.25, 2.5} |
| IQL (Kostrikov et al., 2022) | actor_encoder_factory* | VectorEncoderFactory(hidden_units=[100]) |
| | | VectorEncoderFactory(hidden_units=[30, 30]) |
| | | VectorEncoderFactory(hidden_units=[50, 10]) |
| | critic_encoder_factory* | VectorEncoderFactory(hidden_units=[100]) |
| | | VectorEncoderFactory(hidden_units=[30, 30]) |
| | | VectorEncoderFactory(hidden_units=[50, 10]) |
| | (randomization) class$^{\diamond\diamond}$ | GaussianHead |
| | (randomization) sigma$^\diamond$ | {1.75, 2.0, 2.25, 2.5} |

*Note*: We use the implementations provided by d3rlpy (Seno & Imai, 2021). $*$ and $**$ indicate the parameters and classes of d3rlpy, while $\diamond$ and $\diamond\diamond$ indicate those of our software. For the other parameters that are not specified in the table, we use the default values set by d3rlpy. Note that since we use three different encoders and four different values of sigma, we obtain 12 different policies for both CQL and IQL.

Table 7: Models and parameters used to train the candidate policies in Hopper

| model | parameter | value |
|---|---|---|
| CQL (Kumar et al., 2020) | actor_encoder_factory* | VectorEncoderFactory(hidden_units=[100]) |
| | | VectorEncoderFactory(hidden_units=[30, 30]) |
| | | VectorEncoderFactory(hidden_units=[50, 10]) |
| | critic_encoder_factory* | VectorEncoderFactory(hidden_units=[100]) |
| | | VectorEncoderFactory(hidden_units=[30, 30]) |
| | | VectorEncoderFactory(hidden_units=[50, 10]) |
| | q_func_factory* | MeanQFunctionFactory() |
| | (randomization) class$^{\diamond\diamond}$ | GaussianHead |
| | (randomization) sigma$^{\diamond}$ | {0.5, 1.0, 1.5, 2.0} |
| IQL (Kostrikov et al., 2022) | actor_encoder_factory* | VectorEncoderFactory(hidden_units=[100]) |
| | | VectorEncoderFactory(hidden_units=[30, 30]) |
| | | VectorEncoderFactory(hidden_units=[50, 10]) |
| | critic_encoder_factory* | VectorEncoderFactory(hidden_units=[100]) |
| | | VectorEncoderFactory(hidden_units=[30, 30]) |
| | | VectorEncoderFactory(hidden_units=[50, 10]) |
| | (randomization) class$^{\diamond\diamond}$ | GaussianHead |
| | (randomization) sigma$^{\diamond}$ | {0.5, 1.0, 1.5, 2.0} |

*Note*: We use the implementations provided by d3rlpy (Seno & Imai, 2021). $*$ and $**$ indicate the parameters and classes of d3rlpy, while $\diamond$ and $\diamond\diamond$ indicate those of our software. For the other parameters that are not specified in the table, we use the default values set by d3rlpy. Note that since we use three different encoders and four different values of sigma, we obtain 12 different policies for both CQL and IQL.

Table 8: Models and parameters used to train the candidate policies in Swimmer

| model | parameter | value |
|---|---|---|
| CQL (Kumar et al., 2020) | actor_encoder_factory* | VectorEncoderFactory(hidden_units=[100]) |
| | | VectorEncoderFactory(hidden_units=[30, 30]) |
| | | VectorEncoderFactory(hidden_units=[50, 10]) |
| | critic_encoder_factory* | VectorEncoderFactory(hidden_units=[100]) |
| | | VectorEncoderFactory(hidden_units=[30, 30]) |
| | | VectorEncoderFactory(hidden_units=[50, 10]) |
| | q_func_factory* | MeanQFunctionFactory() |
| | (randomization) class$^{\diamond\diamond}$ | GaussianHead |
| | (randomization) sigma$^{\diamond}$ | {1.0, 1.5, 2.0, 2.5} |
| IQL (Kostrikov et al., 2022) | actor_encoder_factory* | VectorEncoderFactory(hidden_units=[100]) |
| | | VectorEncoderFactory(hidden_units=[30, 30]) |
| | | VectorEncoderFactory(hidden_units=[50, 10]) |
| | critic_encoder_factory* | VectorEncoderFactory(hidden_units=[100]) |
| | | VectorEncoderFactory(hidden_units=[30, 30]) |
| | | VectorEncoderFactory(hidden_units=[50, 10]) |
| | (randomization) class$^{\diamond\diamond}$ | GaussianHead |
| | (randomization) sigma$^{\diamond}$ | {1.0, 1.5, 2.0, 2.5} |

*Note*: We use the implementations provided by d3rlpy (Seno & Imai, 2021). $*$ and $**$ indicate the parameters and classes of d3rlpy, while $\diamond$ and $\diamond\diamond$ indicate those of our software. For the other parameters that are not specified in the table, we use the default values set by d3rlpy. Note that since we use three different encoders and four different values of sigma, we obtain 12 different policies for both CQL and IQL.

Table 9: Models and parameters used to train the candidate policies in CartPole

| model | parameter | value |
|---|---|---|
| CQL (Kumar et al., 2020) | encoder_factory* | VectorEncoderFactory(hidden_units=[100])
VectorEncoderFactory(hidden_units=[30, 30])
VectorEncoderFactory(hidden_units=[50, 10]) |
| | q_func_factory* | MeanQFunctionFactory() |
| | target_update_interval* | 100 |
| | batch_size* | 32 |
| | learning_rate* | 6.25e-5 |
| | (randomization) class$^{\diamond\diamond}$ | EpsilonGreedyHead |
| | (randomization) epsilon$^{\diamond}$ | {0.1, 0.3, 0.5, 0.7} |
| BCQ (Fujimoto et al., 2019) | encoder_factory* | VectorEncoderFactory(hidden_units=[100])
VectorEncoderFactory(hidden_units=[30, 30])
VectorEncoderFactory(hidden_units=[50, 10]) |
| | q_func_factory* | MeanQFunctionFactory() |
| | target_update_interval* | 1000 |
| | batch_size* | 32 |
| | learning_rate* | 1e-3 |
| | action_flexibility | 0.3 |
| | beta | 0.1 |
| | (randomization) class$^{\diamond\diamond}$ | EpsilonGreedyHead |
| | (randomization) epsilon$^{\diamond}$ | {0.1, 0.3, 0.5, 0.7} |

*Note*: We use the implementations provided by d3rlpy (Seno & Imai, 2021). $*$ and $**$ indicate the parameters and classes of d3rlpy, while $\diamond$ and $\diamond\diamond$ indicate those of our software. For the other parameters that are not specified in the table, we use the default values set by d3rlpy. Note that since we use three different encoders and four different values of epsilon, we obtain 12 different policies for both CQL and BCQ.

Table 10: Models and parameters used to train the candidate policies in MountainCar

| model | parameter | value |
|---|---|---|
| CQL (Kumar et al., 2020) | encoder_factory* | VectorEncoderFactory(hidden_units=[100])
VectorEncoderFactory(hidden_units=[30, 30])
VectorEncoderFactory(hidden_units=[50, 10]) |
| | q_func_factory* | MeanQFunctionFactory() |
| | target_update_interval* | 100 |
| | batch_size* | 32 |
| | learning_rate* | 1e-3 |
| | alpha* | 1.0 |
| | (randomization) class$^{\diamond\diamond}$ | EpsilonGreedyHead |
| | (randomization) epsilon$^{\diamond}$ | {0.0, 0.1, 0.2, 0.3} |
| BCQ (Fujimoto et al., 2019) | encoder_factory* | VectorEncoderFactory(hidden_units=[100])
VectorEncoderFactory(hidden_units=[30, 30])
VectorEncoderFactory(hidden_units=[50, 10]) |
| | q_func_factory* | MeanQFunctionFactory() |
| | target_update_interval* | 100 |
| | batch_size* | 32 |
| | learning_rate* | 1e-3 |
| | action_flexibility | 0.3 |
| | beta | 0.01 |
| | (randomization) class$^{\diamond\diamond}$ | EpsilonGreedyHead |
| | (randomization) epsilon$^{\diamond}$ | {0.0, 0.1, 0.2, 0.3} |

*Note*: We use the implementations provided by d3rlpy (Seno & Imai, 2021). $*$ and $**$ indicate the parameters and classes of d3rlpy, while $\diamond$ and $\diamond\diamond$ indicate those of our software. For the other parameters that are not specified in the table, we use the default values set by d3rlpy. Note that since we use three different encoders and four different values of epsilon, we obtain 12 different policies for both CQL and BCQ.

Table 11: Models and parameters used to train the candidate policies in Acrobot

| model | parameter | value |
|---|---|---|
| CQL (Kumar et al., 2020) | encoder_factory* | VectorEncoderFactory(hidden_units=[100]) |
| | | VectorEncoderFactory(hidden_units=[30, 30]) |
| | | VectorEncoderFactory(hidden_units=[50, 10]) |
| | q_func_factory* | MeanQFunctionFactory() |
| | target_update_interval* | 100 |
| | batch_size* | 32 |
| | learning_rate* | 6.25e-5 |
| | (randomization) class$^{\diamond\diamond}$ | EpsilonGreedyHead |
| | (randomization) epsilon$^{\diamond}$ | {0.2, 0.3, 0.4, 0.5} |
| BCQ (Fujimoto et al., 2019) | encoder_factory* | VectorEncoderFactory(hidden_units=[100]) |
| | | VectorEncoderFactory(hidden_units=[30, 30]) |
| | | VectorEncoderFactory(hidden_units=[50, 10]) |
| | q_func_factory* | MeanQFunctionFactory() |
| | target_update_interval* | 1000 |
| | batch_size* | 32 |
| | learning_rate* | 5e-5 |
| | action_flexibility | 0.1 |
| | beta | 0.01 |
| | (randomization) class$^{\diamond\diamond}$ | EpsilonGreedyHead |
| | (randomization) epsilon$^{\diamond}$ | {0.05, 0.10, 0.15, 0.20} |

*Note*: We use the implementations provided by d3rlpy (Seno & Imai, 2021). $*$ and $**$ indicate the parameters and classes of d3rlpy, while $\diamond$ and $\diamond\diamond$ indicate those of our software. For the other parameters that are not specified in the table, we use the default values set by d3rlpy. Note that since we use three different encoders and four different values of epsilon, we obtain 12 different policies for both CQL and BCQ.

Table 12: Models and parameters used in OPE

| model etc. | parameter | value |
|---|---|---|
| FQE (Le et al., 2019) | encoder_factory* | VectorEncoderFactory(hidden_units=[100]) |
| | q_func_factory* | MeanQFunctionFactory() |
| | learning_rate* | 1e-3 |
| DICE (Yang et al., 2020) | (all model parameters)$^{\diamond}$ | (default values) |
| | sigma$^{\diamond}$ | 0.5 |
| continuous OPE | sigma$^{\diamond}$ | 0.5 |

*Note*: We use the FQE implementations provided by d3rlpy (Seno & Imai, 2021) and DICE implementation available in our software. $*$ indicate the parameters of d3rlpy, while $\diamond$ indicate those of our software. sigma is the bandwidth hyperparameter of the kernel function used in DICE and continuous OPE. Note that, for both FQE and DICE, we disable state_scaler for Reacher and enable it for the other enviroments to stabilize the training process.

