# OpenReview forum: "Towards Assessing and Benchmarking Risk-Return Tradeoff of Off-Policy Evaluation"
_ICLR.cc/2024/Conference — ICLR 2024 poster_

### Official Review · Reviewer_n49S · 2023-10-29

**Soundness:** 3 good
**Presentation:** 3 good
**Contribution:** 3 good
**Rating:** 8
**Confidence:** 3

**Summary:**

This paper proposed a new metric for off-policy evaluation, names SharpeRatio@K. This metric measures the risk-return tradeoff and efficiency of policy portfolios formed by an OPE estimator under varying online evaluation budgets (i.e. top-K policies selected by the estimator). Via examples, the paper demonstrates that existing metrics (MSE, regret@1, rank correlation) fail to differentiate policies with different risk-return tradeoffs while the proposed new metric does. The authors evaluated SharpeRatio@K via benchmark experiments using various existing OPE estimators regarding their risk-return tradeoff. The authors also developed open-source software for using the proposed metric.

**Strengths:**

- Overall the paper has a good clarity and is well-organized. I found the examples on over/under-estimation, conservative / high-stakes estimation to be helpful for understanding the benefit of the proposed metric.
- Given that in practice, OPE is more often used as a screening process for selecting top-K policies to deploy in A/B tests, the risk-to-return ratio can be a useful and meaningful metric for comparing OPE estimators. The proposed metric based on Sharpe Ratio to be natural and easy-to-evaluate.
- The benchmark experiments / open-source software provide good evidence that SharpeRatio@K is capable of measuring the risk-to-return efficiency of various OPE estimators, and facilitate the usage of such metric in future research.

**Weaknesses:**

The benchmark results show that SharpeRatio@k can sometime diverge significantly from the conventional metrics in terms of estimator selection. Additional discussion should be added on how practitioners may consolidate the insights given by the different metrics evaluated under certain scenarios.

**Questions:**

- The classic sharpe ratio uses the mean return of the portfolio as the numerator instead of the best policy's return. Is there a particular reason that the best return is used in the proposed metric?

---

> ### Author Response · Authors · 2023-11-13
> **Author Response**
>
> We would like to thank the reviewer for valuable and thoughtful feedback. In particular, we appreciate the supportive comments and acknowledgment of our contributions. We will respond to the key comments and questions below.
>
> > **The classic sharpe ratio uses the mean return of the portfolio as the numerator instead of the best policy's return. Is there a particular reason that the best return is used in the proposed metric?**
>
> This is a great point. We focus on the best policy’s performance in the numerator because only the best policy in the policy portfolio will be finally chosen via A/B testing as a production policy, as illustrated in Figure 1. To define the numerator, we also subtract the behavior policy’s performance ($J(\pi_b)$). By doing so, we can avoid an undesired edge case where an estimator that chooses only policies that are worse than the behavior policy but accidentally has extremely small std (denominator) is evaluated as the best method under our metric. We will clarify this in Section 3.
>
>
> > **The benchmark results show that SharpeRatio@k can sometime diverge significantly from the conventional metrics in terms of estimator selection. Additional discussion should be added on how practitioners may consolidate the insights given by the different metrics evaluated under certain scenarios.**
>
> Thank you for the valuable feedback. The estimator selection results under SharpeRatio@k sometimes diverge from others because SharpeRatio takes the risk into account when evaluating OPE estimators, while other metrics ignore the risk factor. In general, it would be ideal to be able to develop an estimator that performs best in terms of both existing metrics and SharpeRatio. If it is not possible, one should prioritize SharpeRatio@k more when they do not want to take any risk of deploying detrimental policies in the A/B test. When an estimator is preferred by only existing metrics, the estimator is likely to be high-return but also high-risk, so it should be avoided in high-stakes scenarios such as medical applications. We will provide an in-depth discussion about this point in the revision.

---

### Official Review · Reviewer_qCGC · 2023-11-03

**Soundness:** 2 fair
**Presentation:** 2 fair
**Contribution:** 2 fair
**Rating:** 5
**Confidence:** 2

**Summary:**

The paper proposes a new Off-line policy evaluation technique that account for the risk-return tradeoff as usually done in the financial literature when evaluating a portfolio.
Namely the papers proposes a new evaluation metric (SharpeRatio@k) that measure the risk/return tradeoff of OPEs.
The paper is complemented by empirical analysis to evaluate the effectiveness of the method.

**Strengths:**

The paper studies the important problem of offline policy evaluation and propose a risk-aware method for selecting OPE estimators. The idea of incorporating the variance of the estimators seems novel and worth investigating. Moreover the paper is fairly well structured and clearly written.

**Weaknesses:**

My biggest concern is about the technical contribution of the work. While the idea seems novel it also sound quite natural and simple and bears concern about the actual interest that would spark in the community. Moreover the introduction of Sharpe ratio like measure in OPE it seems poorly motivated by the authors who should try to sell better the motivations for the idea

**Questions:**

1) Way is it important to consider the std of the estimators in assessing OPEs? In finance, big swings might prompt the activation of risk measures, and thus one usually prefers loosing some performance points in favour of more stable results. Way is this important in OPE?
2) Is there any provable theoretical advantages of using your method compared to others?
3) Seems like $\hat J(\pi; D)$ was never defined explicitly.

---

> ### Author Response · Authors · 2023-11-13
> **Author Response (1/2)**
>
> We would like to thank the reviewer for valuable and thoughtful feedback. We will respond to the key comments and questions below.
>
> > **My biggest concern is about the technical contribution of the work. While the idea seems novel it also sound quite natural and simple and bears concern about the actual interest that would spark in the community.**
>
> Thank you for the great comment. First, we think that the simplicity of our evaluation metric is actually one of its important advantages. It means that it can be used fairly easily in future research and practice. Reviewer n49S acknowledges that being natural and easy-to-use is one of the main advantages of our metric. Second, a method’s simplicity does not necessarily mean that it is not novel and valuable. The concept of evaluating OPE estimators’ risk-return tradeoff and the use of SharpeRatio as an evaluator of OPE estimators are the ideas that we have never seen in the relevant literature. Besides, the similar types of contributions have been evaluated highly by the relevant community. We can actually list several published papers that share similar types of contributions to ours, i.e., (1) highlighting the problem of existing evaluation protocols and (2) resolving the issues with simple and easy-to-use metrics.
>
> - **(Fu et al., ICLR2021)** (1) This paper identified that an OPE estimator that has a lower estimation error (MSE) does not always align candidate policies better than those with a higher estimation error. (2) To address this gap in evaluating policy alignment in OPE, the paper proposed simple rank correlation and regret to see if OPE estimators align candidate policies well. These metrics are now recognized as important metrics of OPE and are used in e.g., (Tang and Wiens, 2021), (Qin et al, 2022), and (Chang et al., 2022).
>
> - **(Agarwal et al., NeurIPS2021)** (1) This paper identified that taking the naive average of the performance of online RL can be sensitive to variance, particularly when only a handful of trials are allowed due to high computational costs. (2) To enable a more statistically robust evaluation of online RL in such situations, the paper proposed to use the simple average of the middle 50% of trials. This paper was awarded the *Outstanding Paper Award for NeurIPS 2021*.
>
> - **(Chan et al., ICLR2020)** This paper (1) identified that the naive average of the performance of online RL may overlook a critical safety concern. (2) To see if the RL policy is risk-averse or not, the paper proposed to report the simple average of the worst $\alpha$% of trials (referred to as conditional value at risk; CVaR). CVaR is widely used in safe-RL.
>
> **It would thus be great if the reviewer could evaluate our contributions based on their quality, significance, soundness, impact, rather than what the proposed method looks like.** It would also be great if the reviewer could pay attention to another key contribution we have provided, i.e., the development of the open-source software.
>
> ====
>
> (Fu et al., ICLR2021) Justin Fu, Mohammad Norouzi, Ofir Nachum, George Tucker, Ziyu Wang, Alexander Novikov, Mengjiao Yang, Michael R. Zhang, Yutian Chen, Aviral Kumar, Cosmin Paduraru, Sergey Levine, Tom Le Paine. “Benchmarks for Deep Off-Policy Evaluation.”
>
> (Agarwal et al., NeurIPS2021) Rishabh Agarwal, Max Schwarzer, Pablo Samuel Castro, Aaron Courville, Marc G. Bellemare. “Deep Reinforcement Learning at the Edge of the Statistical Precipice.”
>
> (Chan et al., ICLR2020) Stephanie C.Y. Chan, Samuel Fishman, John Canny, Anoop Korattikara, Sergio Guadarrama. “Measuring the Reliability of Reinforcement Learning Algorithms.”
>
> (Tang and Weins, 2021) Shengpu Tang, Jenna Wiens. "Model selection for offline reinforcement learning: Practical considerations for healthcare settings." Machine Learning for Healthcare Conference, 2021.
>
> (Qin et al., 2022) Rongjun Qin, Songyi Gao, Xingyuan Zhang, Zhen Xu, Shengkai Huang, Zewen Li, Weinan Zhang, and Yang Yu. “NeoRL: A Near Real-World Benchmark for Offline Reinforcement Learning.” NeurIPS Datasets & Benchmarks, 2022.
>
> (Chang et al., 2022) Jonathan D. Chang, Kaiwen Wang, Nathan Kallus, Wen Sun. “NeoRL: A Near Real-World Benchmark for Offline Reinforcement Learning” ICML, 2022.

---

> > ### Author Response · Authors · 2023-11-13
> > **Author Response (2/2)**
> >
> > > **the introduction of Sharpe ratio like measure in OPE it seems poorly motivated by the authors who should try to sell better the motivations for the idea**
> >
> > Let us begin with clarifying that SharpeRatio is not a new estimator in OPE, but is a new metric to evaluate the risk-return tradeoff of OPE estimators (in case the reviewer misunderstands what it is). It is novel because no existing metrics consider the risk aspect of OPE estimators. It is practically important to measure the risk in OPE because, given the two-stage policy evaluation procedure in Figure 1, one would often like to avoid deploying detrimental policies in the A/B testing phase. We have provided two example situations in Section 3 where every existing metric (MSE, Regret, and RankCorr) cannot evaluate the risk in OPE since they focus merely on the accuracy of OPE (MSE), the resulting policy alignment (RankCorr), or top-1 policy selection (Regret). Reviewers S1vJ and n49S acknowledge that the toy examples we provided are helpful for understanding the importance of considering the risk-return tradeoff in the evaluation of OPE estimators and advantages of the proposed metric in their review.
> >
> >
> > > **Way is it important to consider the std of the estimators in assessing OPEs?.. Way is this important in OPE?**
> >
> > Let us clarify that the std in the denominator of SharpeRatio is not the deviation of the estimator, but the deviation of the (true) value of policies in the policy portofolio formed by the estimator. This is the key novelty of our metric and it is practically important to measure this because, given the two-stage policy evaluation procedure in Figure 1, we would often like to avoid deploying detrimental policies in the A/B testing phase, which means that we prefer an estimator that can lead to a portfolio that has lower std. No existing metrics can evaluate this risk aspect of OPE estimators because they do not consider the deviation of the values in the policy portfolio like ours. Due to the ability to evaluate the risk via the std term, our metric resolved the issues of existing metrics in the toy examples.
> >
> >
> > > **Is there any provable theoretical advantages of using your method compared to others?**
> >
> > There are no particular theoretical advantages about our metric. We do not think that such a theoretical proof is necessary or possible to compare evaluation metrics that are defined under completely different motivations. For example, how would we be able to prove that the AUC score is better than the F1 score (and vice versa) for the typical classification problem? An important point here is that there is no metric that can evaluate the risk-return tradeoff of OPE estimators in previous research, and our formulation and metric are the first to enable it. We have demonstrated that existing metrics (MSE, Regret, and RankCorr) cannot evaluate the risk aspect of OPE estimators in Section 3 through toy examples. We (and Reviewers S1vJ and n49S) think this is sufficiently useful to motivate our metric and demonstrate its value. We would thus argue that there is no particular need to perform theoretical analysis in our paper.
> >
> > We would appreciate it if the reviewer could clarify what kind of theoretical analysis about the comparison of different evaluation metrics is desired (or even possible) if our response is not satisfactory.
> >
> >
> > > **Seems like $\hat{J}(\pi; \mathcal{D})$ was never defined explicitly.**
> >
> > Thank you for catching. It is an arbitrary OPE estimator. There is no single definition, but examples include (but not limited to) DM, PDIS, DR, and MIS as we described in Section 4.2. We provided rigorous definitions of these estimators using math notations in Appendix A.1. We will clarify this in the revision.

---

> > > ### Author Response · Authors · 2023-11-18
> > >
> > > Dear reviewer qCGC,
> > >
> > > We would again appreciate your valuable and thoughtful review. Since the deadline for the author-reviewer discussion period is approaching, it would be great to have feedback on whether our response addresses the concerns raised in the initial review.
> > >
> > > Thank you,
> > > authors

---

### Official Review · Reviewer_S1vJ · 2023-11-04

**Soundness:** 4 excellent
**Presentation:** 3 good
**Contribution:** 4 excellent
**Rating:** 8
**Confidence:** 3

**Summary:**

This paper studies the risk-return tradeoff between different OPE evaluations. Existing methods evaluate the superiority of an OPE estimator via various “accuracy” measures. However, the paper argues that merely looking at the accuracy may not be sufficient as two OPEs with similar accuracies may have different risk implications in practice.

To fix this issue, the paper proposes to use concepts from portfolio evaluation in finance and develops a new metric called SharpeRatio@k. This metric helps distinguish between conservative and high-stakes OPE estimators. The key idea behind this is to regard the set of top-k candidate policies selected by an OPE estimator as its “policy portfolio”. The paper constructs a policy portfolio that is “efficient”, i,e, it contains policies that improve the performance of behavior policy without including poorly performing policies. Finally, the paper evaluates typical OPE estimators using the proposed metric using a number of continuous control benchmarks.

**Strengths:**

– I think the main strength of the paper lies in identifying how the portfolio evaluation concepts in finance can be applied to evaluation of OPEs. I’m not an expert in finance, and can’t speak to the novelty of the idea, but assuming it is novel, this certainly sounds interesting to me.

– The paper is well-organized, is a pleasure to read and explains difficult concepts well enough.

– I also really liked the use of toy examples throughout the paper to drive home the key concepts.

– Experiments section is thorough; it compares several of commonly used OPE estimators and also identifies several directions for future OPE research.

**Weaknesses:**

– I felt the figures and toy examples could use more explanation. For example, it wasn’t clear in Fig 2 what the red dots are, black dots are and how to interpret the axes.

– Related work: In the section on “Risk Measures and risk-Return Tradeoff in Statistics and Finance”, the paper discusses the Sharpe,1998 paper in length (relevant for the Sharpe ratio used in current paper). However, there are only two other related works identified in this entire section. I’m not an expert in this domain but I suspect there has to be more prior research done in this space – and if that is true, the existing related work section seems a little thin.

– Other minor suggestions:

1.  In the abstract, the sentence “We first demonstrate, …” is a bit too long and unwieldy and hard to understand easily.

2.  Contributions paragraph talks about “top-k policy deployment” without any prior context on what that means. Difficult to follow.

**Questions:**

– In description of Fig 2, I found the sentence “X underestimates the performance… Y overestimates ” confusing – I was wondering if it should be the other way around. If the current sentence has to be true, then my understanding of the interpretation is that black policies suffer an underestiation (i.e. even though x-axis value is high, y-axis value is lower than ground truth) and so the top-k ends up picking next best policies. Is this correct or am I missing something?

– top-k policies are considered in several resource allocation setups where there are limited resources and some index is computed so that resource can be allocated to the top k indexes
(For e.g. [1] “Restless Bandits: Activity Allocation in Changing World”, P Whittle; [2] https://arxiv.org/abs/2110.02128). Can the method be using for designing robust top-k selection/resource allocation policies in this context? I also wonder how it compares to existing robustness in bandits work such as [3] https://arxiv.org/abs/2107.01689 and whether those ideas can be applied to the OPE evaluation setting discussed in the current paper.

---

> ### Author Response · Authors · 2023-11-13
> **Author Response**
>
> We would like to thank the reviewer for valuable and thoughtful feedback. We will improve the presentation of the paper following the reviewer's suggestions. We will respond to the key comments and questions below.
>
> > **it wasn’t clear in Fig 2 what the red dots are, black dots are and how to interpret the axes.**
>
> > **I found the sentence “X underestimates the performance… Y overestimates ” confusing – I was wondering if it should be the other way around. If the current sentence has to be true, then my understanding of the interpretation is that black policies suffer an underestiation (i.e. even though x-axis value is high, y-axis value is lower than ground truth) and so the top-k ends up picking next best policies. Is this correct or am I missing something?**
>
> Thank you for catching the ambiguity. The red dots indicate the policies whose values are estimated correctly by the estimator, while the black dots indicate the policies whose values are not correctly estimated.
>
> Your understanding about estimator X (“my understanding of .. next best policies”) is correct indeed, and the three points in the shaded region are chosen as the top-$k$ policies under estimator X (these three policies have higher estimated policy value (y-axis) than others).
>
> On the other hand, estimator Y substantially overestimates the value of black policies. As a result, one red policy (the rightmost one) and two black policies are chosen as the top-$k$ policies (policy portfolio) under this estimator, which is considered risky because the true values of these black policies are quite low.
>
>
> > **Related work: In .. “Risk Measures and risk-Return Tradeoff in Statistics and Finance”, the paper discusses the Sharpe,1998 paper in length .. However, there are only two other related works .. I’m not an expert in this domain but I suspect there has to be more prior research done in this space – and if that is true, the existing related work section seems a little thin.**
>
> Thank you for the valuable comment. Indeed, there are several variants of risk measures used in finance such as the information ratio, value at risk (VaR), marginal VaR, and entropic VaR (some of them have been used in bandits and RL). We will add comments about these relevant metrics in the section and keep improving the writing of the paper.
>
>
> >  **top-k policies are considered in several resource allocation setups where there are limited resources and some index is computed so that resource can be allocated to the top k indexes .. Can the method be using for designing robust top-k selection/resource allocation policies in this context? I also wonder how it compares to existing robustness in bandits work such as .. and whether those ideas can be applied to the OPE evaluation setting discussed in the current paper.**
>
> We agree that the high-level concept of the risk-return tradeoff can be applied to the problem of resource allocation and restless bandit as suggested by the reviewer. A notable difference is that we used the concept to define a new metric to evaluate OPE estimators while the same concept can be useful to define a new performance measure of a bandit policy in the field of resource allocation and restless bandit. We can actually find some similar attempts such as the risk-aversion bandit (Sani et al. 2012). Even though this is an independent interest, we thank the reviewer for sharing this interesting thought.
>
> (Sani et al. 2012) Amir Sani, Alessandro Lazaric, and Rémi Munos. Risk-Aversion in Multi-armed Bandits. NuerIPS2012.

---

### Official Review · Reviewer_wVqE · 2023-11-08

**Soundness:** 3 good
**Presentation:** 3 good
**Contribution:** 2 fair
**Rating:** 5
**Confidence:** 3

**Summary:**

This paper is concerned with defining an appropriate measure for off-policy evaluation. The authors argue that commonly used metrics such as MSE and rank correlation are insufficient because they fail to properly account for risk/return tradeoffs. To remedy this it’s proposed that ideas from portfolio optimization be used. In particular, the authors propose the use of the Sharpe ratio to measure efficiency. A number of empirical results are provided which demonstrate the properties of the Sharpe ratio @ k metric, and compare it to other metrics such as rank correlation and regret.

**Strengths:**

This paper tackles an important problem–off policy evaluation is a critical aspect of deployment of RL systems in many real world contexts. The authors' proposal to use ideas from portfolio optimization is an interesting one, and the proposal to use Sharpe @ k is both intuitive and simple. The authors do a nice job of evaluating their work empirically and demonstrating the properties of the proposal.

**Weaknesses:**

My biggest issue with this paper is that the work is very limited in scope which limits the benefit to the larger community. While the proposal to use ideas from portfolio theory is interesting, the authors focus on a fairly simple definition and don't describe the properties and behavior of the proposed approach theoretically. It would be useful if the authors described the proposal in slightly more generality. For example, are there rank-based analogs of the current approach? It would also be useful if there was a full discussion of the necessary assumptions/conditions for Sharpe@k to be applicable in a real world setting. It would also be useful if the authors highlighted cases where current metrics could be preferable. In general, it would seem that the appropriateness of a given evaluation metric is entirely task dependent, a discussion of this could be useful.

**Questions:**

It would be useful if the authors could give a description of the settings in which one would prefer Sharpe@k generally. Also, would be useful if the authors could characterize the necessary assumptions on the reward distributions.

---

> ### Author Response · Authors · 2023-11-13
> **Author Response (1/2)**
>
> We would like to thank the reviewer for valuable and thoughtful feedback. We will respond to the key comments and questions below.
>
> > **My biggest issue with this paper is that the work is very limited in scope which limits the benefit to the larger community.**
>
> Thank you for the great point. First, **off-policy evaluation (OPE) of bandits and RL is an established and broad research area and is widely recognized at top-tier conferences like ICLR, ICML, and NeurIPS** (more than 15 relevant papers were accepted to these venues in each of 2021-2023). Second, our work highlighted the issue of the existing evaluation protocol of OPE estimators and proposed a new framework to evaluate their risk-return tradeoff. In Section 4, we have indeed demonstrated that estimators that were considered pretty basic can be the most effective method under our metric. This would have a potential to change the way we evaluate the effectiveness of estimators and develop new estimators in the future of the entire OPE research community as we discussed in Section 5. It would also be possible that our concept of evaluating the risk-return tradeoff will lead to a development of new evaluation metrics for other fields such as fair machine learning, recommender systems, and treatment effect estimation. Thus, we believe that our work is not limited in scope but provides a whole new perspective regarding the evaluation of OPE and potentially many other fields. We will add this discussion in the revision.
>
>
> > **While the proposal to use ideas from portfolio theory is interesting, the authors focus on a fairly simple definition**
>
> Thank you for the great comment. First, the simplicity of our evaluation metric is actually one of its important advantages. It means that it can be used fairly easily in future research and practice. Reviewer n49S also acknowledges that being natural and easy-to-use is one of the main advantages of our metric. Second, a method’s simplicity does not necessarily mean that it is not novel and valuable. The concept of evaluating OPE estimators’ risk-return tradeoff and the use of SharpeRatio as an evaluator of OPE estimators are the ideas that we have never seen in the relevant literature. Besides, the similar types of contributions have been evaluated highly by the relevant community. For example, (Fu et al., ICLR2021) proposed the rank correlation and regret as simple evaluation metrics for OPE. (Agarwal et al., NeurIPS2021) proposed the average of the middle 50% of trials as the online RL performance. The values of these papers have been recognized widely because they highlight the issues of existing evaluation protocols and develop a simple metric to deal with them. So is our work, and we believe that the reviewer’s comment actually highlights one of the most important pros of our metric. **It would thus be great if the reviewer could evaluate our contributions based on their quality, significance, soundness, impact, rather than what the proposed method looks like.**
>
> ===
>
> (Fu et al., ICLR2021) Justin Fu, Mohammad Norouzi, Ofir Nachum, George Tucker, Ziyu Wang, Alexander Novikov, Mengjiao Yang, Michael R. Zhang, Yutian Chen, Aviral Kumar, Cosmin Paduraru, Sergey Levine, Tom Le Paine. “Benchmarks for Deep Off-Policy Evaluation.” ICLR, 2021.
>
> (Agarwal et al., NeurIPS2021) Rishabh Agarwal, Max Schwarzer, Pablo Samuel Castro, Aaron Courville, Marc G. Bellemare. “Deep Reinforcement Learning at the Edge of the Statistical Precipice.” NeurIPS, 2021.

---

> > ### Author Response · Authors · 2023-11-13
> > **Author Response (2/2)**
> >
> > > **It would also be useful if the authors highlighted cases where current metrics could be preferable. In general, it would seem that the appropriateness of a given evaluation metric is entirely task dependent, a discussion of this could be useful.**
> >
> > > **It would be useful if the authors could give a description of the settings in which one would prefer Sharpe@k generally.**
> >
> > We are absolutely happy to elaborate. One would prefer SharpeRatio@k when they care about the risk of deploying detrimental policies in the subsequent A/B test given the two-stage evaluation procedure depicted in Figure 1. We have provided two example situations in Section 3 where every existing metric (MSE, Regret, and RankCorr) cannot evaluate the risk since they focus merely on the accuracy of OPE (MSE), the resulting policy alignment (RankCorr), or top-1 policy selection (Regret). Of course, one can use ShapeRatio and some existing metrics simultaneously depending on their needs. But our point is that it was not possible to evaluate the risk-return tradeoff of estimators without the development of our metric since no existing metrics enable it as we have shown via toy examples and benchmark experiments. Reviewers S1vJ and n49S acknowledge that the toy examples we provided are helpful for understanding the importance of considering the risk-return tradeoff in the evaluation of OPE estimators and advantages of the proposed metric in their review.
> >
> >
> > > **It would also be useful if there was a full discussion of the necessary assumptions/conditions for Sharpe@k to be applicable in a real world setting.**
> >
> > > **Also, would be useful if the authors could characterize the necessary assumptions on the reward distributions.**
> >
> > The only requirement about reward distribution is that its expectation and variance should exist. Most well-known distributions satisfy this (a rare exception is the Cauchy distribution). Note that this is necessary to define not only SharpeRatio but also existing metrics such as MSE, Regret, and RankCorr. There are no other particular requirements needed for the feasibility of our and existing metrics. We will clarify this in the revision.

---

> > > ### Author Response · Authors · 2023-11-18
> > >
> > > Dear reviewer wVqE,
> > >
> > > We would again appreciate your valuable and thoughtful review. Since the deadline for the author-reviewer discussion period is approaching, it would be great to have feedback on whether our response addresses the concerns raised in the initial review.
> > >
> > > Thank you,
> > > authors

---

### Meta-Review · Area_Chair_EcP7 · 2023-12-09

**Metareview:**

Off-policy evaluation (OPE) has become an integral part of industrial RL applications, such as recommendation and search systems. A typical offline-to-online workflow involves offline experimentation, wherein a set of candidate policies is identified with OPE; these candidates are then winnowed down to a single deployed policy via online A/B testing. In this workflow, the choice of OPE estimator is critically important, and this depends on how estimator quality is measured. To date, most studies use MSE or rank correlation (w.r.t. some ground truth, which could come from A/B testing). Unfortunately, neither metric takes into account the _risk-return_ trade-off. Inspired by _portfolio optimization_ in finance, this paper proposes a new metric, _SharpeRatio@k_, which measures the risk-return trade-off. Via comprehensive experiments, the paper demonstrates that SharpeRatio@k can distinguish between estimators, and can identify the best estimator for selecting a portfolio of candidate policies for A/B testing. The proposed metric is implemented in open source software.

The reviewers agree that the paper is well written and organized, with nice illustrative examples to help the reader. The proposed method seems novel (for OPE), and are no (substantiated) questions about the work's soundness. The empirical evaluation seems thorough.

The main critiques are that the proposed method is simple, from a technical perspective, and that it might have limited impact to the ICLR community. Regarding simplicity, I'm of the opinion that simple solutions can be quite effective, and contributions should be measured by their potential impact instead of just their technical innovation. I also disagree with the assertion that the impact will be limited; OPE is becoming an integral part industrial ML systems, and has been an active area of research for the past decade or so.

Other critiques about motivation, expanding discussion of related work, and explanation of empirical results, seem like they can be handled in a revision. I am not concerned about these. (That said, I encourage the authors to use these questions and misunderstandings to identify parts of the paper that need clarification.)

**Justification For Why Not Higher Score:**

Based on the reviews and my skimming, it seems like a good paper, but not _amazing_. I think it will be of interest to the ICLR community, but I'm not sure it should have a spotlight talk (let alone a full talk).

**Justification For Why Not Lower Score:**

As I said in my review, I don't think any of the criticisms warrant rejection. That said, maybe I have missed some aspect

---

### Decision · Program_Chairs · 2024-01-16

Accept (poster)